# A Comprehensive Review of Machine Learning Used to Combat COVID-19

**DOI:** 10.3390/diagnostics12081853

**Published:** 2022-07-31

**Authors:** Rahul Gomes, Connor Kamrowski, Jordan Langlois, Papia Rozario, Ian Dircks, Keegan Grottodden, Matthew Martinez, Wei Zhong Tee, Kyle Sargeant, Corbin LaFleur, Mitchell Haley

**Affiliations:** 1Department of Computer Science, University of Wisconsin-Eau Claire, Eau Claire, WI 54701, USA; kamrowcj0380@uwec.edu (C.K.); langloij0638@uwec.edu (J.L.); dircksiw2283@uwec.edu (I.D.); grottokj2390@uwec.edu (K.G.); martinma5716@uwec.edu (M.M.); teewz1076@uwec.edu (W.Z.T.); sargeakb9348@uwec.edu (K.S.); lafleucj6782@uwec.edu (C.L.); haleymj5781@uwec.edu (M.H.); 2Department of Geography and Anthropology, University of Wisconsin-Eau Claire, Eau Claire, WI 54701, USA; rozaripf@uwec.edu

**Keywords:** COVID-19 prognosis, deep learning, machine learning, CT scan, X-rays

## Abstract

Coronavirus disease (COVID-19) has had a significant impact on global health since the start of the pandemic in 2019. As of June 2022, over 539 million cases have been confirmed worldwide with over 6.3 million deaths as a result. Artificial Intelligence (AI) solutions such as machine learning and deep learning have played a major part in this pandemic for the diagnosis and treatment of COVID-19. In this research, we review these modern tools deployed to solve a variety of complex problems. We explore research that focused on analyzing medical images using AI models for identification, classification, and tissue segmentation of the disease. We also explore prognostic models that were developed to predict health outcomes and optimize the allocation of scarce medical resources. Longitudinal studies were conducted to better understand COVID-19 and its effects on patients over a period of time. This comprehensive review of the different AI methods and modeling efforts will shed light on the role that AI has played and what path it intends to take in the fight against COVID-19.

## 1. Introduction

COVID-19 is a viral infectious disease caused by SARS-CoV-2 that affects the respiratory system of infected persons and can range from mild to severe symptoms. The spread of the disease is predominantly through small aerosol particles that are expelled when coughing, sneezing, breathing, or talking. Comorbidities such as cardiovascular disease, diabetes, chronic respiratory disease, or cancer can increase the chances of developing serious illness or medical complications [1]. COVID-19 is primarily diagnosed using reverse transcription polymerase chain reaction (RT-PCR) tests which includes specific detection of the sequences for the nucleocapsid, envelope, spike, and RNA-dependent RNA polymerase proteins of the virus [1]. These tests have high specificity, but their sensitivity rates could be as low as 60–70% [2]. The use of artificial intelligence (AI) in this area has made significant contributions in this regard to identify, forecast, and treat COVID-19. There are several forms of medical data ranging from text to images in the form of X-rays or CT scans. Text data can include patient health and several factors being recorded in the hospital after COVID-19 diagnosis. Radiation information can be used to diagnose COVID-19 and also study the progression of this disease. Machine Learning algorithms are equipped with the potential to detect hidden trends and aid a clinician in the decision-making process which can save countless lives and utilize hospital resources in an optimal way. However, to train such accurate models, machine learning algorithms need access to huge datasets.

The use of machine learning models as diagnostic and prognostic tools helps to increase both accuracy and speed while ensuring desired patient outcomes. Prior to COVID-19, researchers and clinicians have had tremendous success in using machine learning to identify and classify chest radiographs for multiple applications including the diagnosis of pneumonia [3,4]. Machine learning algorithms like Naïve Bayes, K-nearest neighbor and Support Vector Machines(SVM) were all optimized with the best features. Machine learning has also been applied to predict symptomatic intracranial hemorrhaging from brain CTs and clinical variables in patients suffering from acute ischemic stroke [5]. Results were used to determine whether to administer intravenous thrombolysis (tPA) which can result in the dissolution of the blood clot or can lead to complications from intracranial hemorrhaging. Machine learning also finds application in the identification of liver tumor boundaries in CT imaging [6]. Machine learning techniques like AdaBoost allowed a model to learn diverse intensity distributions of the tumor regions and then use those results for developing a detection algorithm. Machine learning also finds applications in fields such as urinary bladder cancer staging in CT urography [7]. Various automated segmentation techniques can detect bladder lesions and then extract both texture and morphological features for classification.

In this review, we summarize some of the machine learning architectures that have been successful at an accurate diagnosis of COVID-19 using primarily CT imaging datasets. Based on the existing literature presented in research similar to [8,9], we primarily focused on machine learning models developed and published in the years 2021, and 2022. Using a combination of standard keywords in Google Scholar such as “COVID-19”, “Machine Learning”, “Deep Learning”, “Diagnostic COVID-19”, “Longitudinal COVID-19”, and “COVID-19 prognosis”, we report 108 of the most relevant papers that played a significant role in the shaping the research. We found 10 papers from 2020, 47 from 2021, and 51 from 2022. Figure 1 shows a summary of this research based on country, as well as publishers. This paper’s main contribution is as follows:Summary of the current AI applications in the COVID-19 setting ranging diagnostic, to prognostic models.In diagnostic models, classification and segmentation approaches have been presented. The diagnostic models are primarily focused on deep learning while the prognostic and longitudinal study explores machine learning.Longitudinal studies were reported in order to better understand how COVID-19 affected patients over time. This involved using serial medical imaging data that showed disease progression at different points in time. This gave researchers the ability to track disease severity and improve classification and segmentation models.The development of prognostic models allowed researchers to diagnose COVID-19 more accurately at different disease stages while providing critical information on a patient’s prognosis. Some of the patient outcomes that could be predicted included: Mortality, Hospital admission, Intensive care unit admission, and Hospital length of stay These predictors also provided valuable information to healthcare systems for the allocation of scarce medical resources.

The remaining article is organized as follows. Section 2 looks at common machine learning techniques identified in this research focusing specifically on deep learning which was extensively used in COVID-19 diagnosis using primary CT images and radiographs complemented with other clinical variables. Section 3 presents the publicly identified datasets that were used by authors for developing machine learning methodologies. Section 4 looks primarily at the use of CT imaging and X-rays for COVID-19 diagnosis. Section 5 looks at research that explores prognostic models as well as studies longitudinal data for monitoring patients suffering from COVID-19. Finally, Section 6 presents the conclusion from this review.

## 2. Machine Architectures Used in COVID-19 Diagnosis

Machine learning refers to the process of using data for generating a model followed by using that model to make predictions. The most commonly used Maximum Likelihood Classification involves the calculation of the probability that a given feature belongs to a specific class label [10,11]. Datasets used for training contains pre-selected feature values that correspond to these classes which in our case could be either ‘normal’ or ‘COVID-19’. Classification is done by determining two values for the classes, the mean and the covariance which is then used in a normal distribution for probability estimation. Each record/data point will be assigned a probability for every class and the highest probability class is selected for the data point. For COVID-19 diagnosis, the commonly used machine learning models identified included random forests, and support vector machines (SVM).

### 2.1. Machine Learning Models

Random Forests are an ensemble learning method of machine learning that performs classification, regression, and other tasks through the use of decision trees. For the $K_{th}$ tree, a random vector is generated that is independent of the past vectors but with the same distribution. From this random vector and the training set, a tree is grown, resulting in a classifier. Thus, a random forest classifier consists of a large collection of tree-structured classifiers and each tree casts a vote for the most popular class [12]. The resiliency of this system allows for the addition of more trees to not induce over-fitting. Random forests converge towards a limited generalization error value as the number of trees increases. The accuracy of this method is dependent on the strength of the tree classifiers along with the correlation between them. SVM is a supervised learning method in machine learning that is used for classification, regression, and outlier detection. Classification can be either binary or multi-class depending on the dataset. The dataset can be described as *p*-dimensional vectors which can be separated by a *p*—1-dimensional hyperplane called a linear classifier. A hyperplane is a subspace whose dimension is one less than that of its ambient space. The optimal hyperplane is the linear decision function with the maximum margins between the vectors of the two classes that are being classified [13]. Figure 2 shows how hyperplanes are used to separate class labels. These vectors that are on the margin can be described as support vectors. To create this separation between the two, these data are transformed into a higher dimensional space that makes classification easier.

### 2.2. Deep Neural Networks

A deep neural network (DNN) is a type of machine learning which uses multiple consecutive hidden layers between the input and output. These hidden layers are used to extract and process the information received, making weighted calculations. When a deep neural network is trained, the weights are adjusted automatically to more closely match labels on manually identified data. The image of a DNN is shown in Figure 3. Here the hidden layers are represented by blue, while yellow represents input and orange represents output.

### 2.3. Convolutional Neural Networks

Convolutional neural networks, or CNNs, are deep learning algorithms useful for image processing. A CNN is overall a black box technique, where the input and output are the only parts we can observe. The internal workings of a CNN can be understood in theory, but we cannot easily discover the internal workings of a trained CNN. This can be a setback, as we do not truly know what part of an image the model considers when making the classification. However, high accuracy on a large variety of testing images serves as proof that a model is performing well. Segmentation networks are slightly superior to classification networks in this regard, as the output image shows what the model is seeing. To extract features from an image, the CNN utilizes kernels. These kernels scan over every pixel of the image (for a stride of one), and a kernel size larger than 2×2 will consider a matrix including three neighboring pixels. A larger kernel size considers more surrounding pixels. For a kernel size of 3×3, it also views the eight surrounding pixels. A kernel utilizes a weighted matrix to interpret the pixel and send it to the feature map. Each convolution outputs a full feature map from the input image. The size of this feature map can change based on the stride used—a stride of one will not change the image size, but a stride of two will halve the size of the image. Stride refers to how far the kernel moves, so a stride greater than one will skip half of the pixels to create the feature map. This is used for downsampling, and it is useful for increasing the computational efficiency of the model. CNNs use multiple layers of convolutions in order to pick up on different levels of features. Earlier layers pick up low-level features, such as lines or curves in the image. Later (or deeper) layers begin to pick up on the high-level features of the image, which are specific details in the image. For example, one low-level feature of a face would be the curve of the forehead. A higher level feature could be the eyes or something equally specific to the image. Besides being used to primarily process images, CNNs have also found a wide variety of applications in other domains like bioinformatics and cybersecurity among others. In cybersecurity research such as [14], the authors present how features can be represented in a matrix format for malware detection applications using CNN. Authors in [15] report the applicability of using CNNs for DNA sequence interpretation. CNNs have also been applied for named entity recognition of biomedical text [16].

The input of a CNN—whether the task is classification or segmentation—is an image. When a CNN model is created, the input shape is specified, and cannot be changed without retraining an entirely new model. If the model is designed to take an input image of shape 256 × 256, an image of shape 128 × 128 must be resized. Preprocessing includes a step in which the images are resized to fit the determined input shape.

The output of CNN defines the differences between model tasks. A classification model, such as ResNet or MobileNet, will provide a prediction of which class the entire image belongs to. This comes as a probability for each class—for two classes, there will be two numbers, each between 0 and 1. A simple binary classification task decides which of the two classes the image belongs to. For COVID-19 detection, these models are often used to classify images as either positive or negative. Segmentation models, on the other hand, will predict the class of every separate pixel of an image. This generates the predicted mask.

#### 2.3.1. UNet

The UNet model [17] architecture shown in Figure 4 is a 2D convolutional neural network that was designed to give precise segmentation given fewer training images. The model is also noted for being very efficient, generating a segmentation of a 256×256 image in a matter of seconds. This architecture follows a contracting path, followed by an expansive path. The diagram of this architecture forms a U-shape, hence the name UNet. The contracting path (left) uses a traditional CNN shape to extract the information while reducing the size of the image. The model uses blocks that contain two 3×3 convolutional layers, a ReLU function, and then a 2×2 Max Pooling layer. The Max Pooling layer uses a stride of 2, which performs downsampling. After each block, the number of feature channels is doubled. The left and right paths of the UNet architecture match such that after each downsample, a cropped version of the feature map is passed on to the respective block during upsampling. The expansive path (right) consists of the mentioned upsampling of the feature map, a 2×2 convolution, a concatenation with the downsample’s respective feature map, two 3×3 convolutions, and finally a ReLU. The 2×2 convolution serves to halve the number of feature channels on the map.

#### 2.3.2. VGG Frameworks

The visual geometry group (VGG) model [18] shown in Figure 5 is a basic deep 2D convolutional neural network architecture. The general structure of a VGG model uses many blocks of convolutional layers, followed by three fully connected layers for the task of classification. The convolutional layers use a kernel size of 3×3, with a convolutional stride of 1 pixel, with a standard input size of 224×224 pixels. Each convolutional block contains a couple of convolutional layers and ends with a pooling layer. This pooling layer is used to reduce the dimensions of the image before moving to the next block. The fully connected layers use the ReLU activation, and the final layer uses the SoftMax activation to predict the output. There are 4096 channels in the first two layers, while the channel in the last layer is equal to the number of output classes. For the ImageNet database (which VGG was created for), there are 1000 channels for each of the 1000 image classes. There are a few types of VGG architectures, denoted by the depth of the layers—VGG16 uses 16 neural network layers (13 convolutional with three fully connected), while VGG19 uses 19.

#### 2.3.3. Residual Networks

The ResNet architecture [19] is a 2D convolutional neural network that uses identity shortcut connections to skip a number of weighted layers in the network. The basic structure of ResNet shown in Figure 6 uses 34 weighted layers, consisting of 33 convolutional layers and only one fully connected layer for classification. The convolutional layers use a 3×3 kernel, and down sampling occurs directly by certain convolutional layers using a stride of 2. ResNet was created to allow a deeper model while avoiding the vanishing gradient problem, which occurs when the weights in deeper layers of the model are not significantly impacted during training. Because of the use of the ReLU activation, along with the residual layers, ResNet is able to use 34 total layers, while this is a lot of layers, it uses fewer filters and is much less complex than similar architectures. The baseline ResNet model only has 18% of the FLOPs seen in the VGG19 architecture.

#### 2.3.4. MobileNet

The MobileNet architecture [20] is a 2D convolutional neural network designed for the purpose of mobile vision applications, and as such, the model is very lightweight. The MobileNet model allows the programmer to trade between latency and accuracy based on the requirements of the application. This is done through the alteration of the two hyper-parameters width multiplier and resolution multiplier. The width multiplier can be used to decrease the number of input channels and output channels by the same factor. This is similar for the resolution multiplier, where the resolution of the input image (and every layer thereafter) is reduced by the same factor. The basic structure of MobileNet uses depthwise separable convolutions, a form of factorized convolutions as shown in Figure 7. These factorize standard convolutions into depthwise and 1×1 pointwise convolutions. Where a standard convolution would filter and combine the inputs in one step, the depthwise convolution applies a single filter for each input channel, and the pointwise convolution combines those outputs. Each layer—both depthwise and pointwise—of the model is followed by batch normalization and a ReLU nonlinearity. To downsample the image, a stride is set in the depthwise convolutions as well as the first convolutional layer. At the end of the model, there is an average pool to reduce the spatial resolution, followed by a fully connected layer using SoftMax classification.

#### 2.3.5. DenseNet

The DenseNet architecture [21] is a 2D convolutional neural network which, similar to ResNet, utilizes identity shortcut connections. In contrast to ResNet, each layer in DenseNet is connected to all of the subsequent layers as shown in Figure 8. This makes full use of the benefits of these connections, and the authors propose that this promotes feature reuse. This makes the architecture very parameter efficient. The design of DenseNet uses two types of blocks—transition layers and dense blocks. A dense block consists of multiple convolutions in a stack, and the output of a layer is directly connected to every other convolution which comes after it. In a dense block containing five convolutions, the fifth layer will take in the outputs of the four previous layers. Each dense block is separated by transition layers. These layers use both a convolution and pooling layer in order to downsample the feature maps. The last step of the model contains a 7×7 global average pool and a fully connected layer with SoftMax for classification. Each variation of DenseNet uses four dense blocks and three transition layers, followed by the classification layer. More computationally intensive (and better performing) DenseNet architectures are created by increasing the number of feature maps in the later dense blocks.

The choice of deep learning algorithms is heavily influenced by the data being processed at any time. For example, deeper models (models with many layers), can be more accurate but often suffer from the vanishing gradient problem. This issue can be resolved by using Residual Networks however, they are computationally expensive. Lightweight models, like ShuffleNet [23] utilizing depthwise separable convolutions, can be very effective but due to limited training parameters, can reduce accuracy. It has also been observed that attention UNet models provide higher accuracy than traditional UNet models [24] but these models could require longer training times. We observe through our reviews that based on the data being analyzed authors have reported a mix of different models being better at classification or segmentation.

#### 2.3.6. Recurrent Neural Networks

For longitudinal or time series data, a simple CNN is not sufficient. It can capture the data points, but it does not have the tools to view them as a series of events. This is desirable for applications such as prognostic models, where the progression of a disease over time is important information in determining the patient’s outcome. This is where the recurrent neural networks (RNNs) step in. An efficient version of the RNN called the long short-term memory (LSTM) concept is used. It is suited for time series predictions of variable durations and considers previous information by using multiple blocks which each consist of forgetting, input, and output gates.

The forget gate gives attention to the right information while ignoring irrelevant information. A sigmoid function is used, assigning an importance value between zero and one to part of the previous output. Necessary data is closer to one, and unnecessary data is closer to zero. The input gate takes the current state and previously hidden state and passes this to a second sigmoid function. This again assigns a value between zero and one, but values closer to zero are now considered important while values closer to one are not. The current state and hidden state are also passed through a tanh function, which regulates the network by keeping values between −1 and 1. The tanh and sigmoid outputs are then multiplied. The use of the output gate is to output a new hidden state. The previous hidden state and the input are passed to a sigmoid function, and the new cell state is passed to a tanh function. This is multiplied with the sigmoid to determine which information was relevant and outputs the hidden state. This creates the final output of the entire cell, which is the new cell state, and the new hidden state, which passes along to the next time step.

By creating a chain of these memory blocks, the LSTM can retain short-term time series information for a long period of time. This is an improvement on the basic recurrent neural network design, which can only retain short-term time series information for a short while. A complete example of an LSTM block is shown in Figure 9.

## 3. Datasets for Research

The datasets included in this review consisted of both public and private COVID-19 datasets that contained Computer Tomography, X-ray, and Ultrasound imaging. The size of these datasets varied widely with one consisting only of 20 images while another one of the largest datasets consisted of over 500,000 images. Most of the datasets that were found came from studies that were more focused on diagnostic imaging and analysis of COVID-19. Prognostic and Longitudinal studies only included 6 public datasets, while many of them were private. Public datasets may lack critical demographic, clinical and patient outcome data associated with imaging data. Access to private datasets through research institutions and healthcare systems is more difficult but would allow for greater control over where the information is coming from along with the types of information that are available to researchers. The most commonly available public repositories for these public datasets were found to be Kaggle and GitHub. Table 1 shows the name of the publicly available COVID-19 dataset along with the modality in which the imaging was conducted. The size of the datasets in a number of images is also included along with a link to the dataset.

## 4. COVID-19 Prediction Using Deep Learning

COVID-19 has tremendously impacted patients and medical systems globally and deep learning has been extensively used to create COVID-19 prediction models. Figure 10 highlights some of the commonly used models by researchers. Much of the datasets used were CT scans and chest radiographs. The research was focused on both classification and segmentation to identify patterns during diagnosis. The following section contains a discussion on variants of deep learning models along with the machine learning approach utilized so far.

### 4.1. Diagnosis Using Chest Radiographs

In [53], the authors proposed CovNNet, a custom convolutional neural network, to detect COVID-19 in CXR. The second approach for the model used a fuzzy preprocessing step, in which fuzzy and normal CXR are fed into separate CovNNet models. The CovNNet architecture consists of three convolutional blocks, which contain a convolutional layer with ReLU activation followed by a max pooling layer. These features are combined, and then the model feeds to a multi-layer perceptron, which classifies the input as either COVID-19 or non-COVID-19. This second approach has the highest accuracy and time requirement of 80.9% and 50 s, respectively. The dataset used contained a CXR of 64 COVID-19 positive patients and 57 patients with interstitial pneumonia. In [54], the authors proposed a join-fusion model which combines clinical data and CXR to increase the accuracy of the final prediction. CXR images are fed into EfficientNetB7 (which uses the ImageNet weights) which feeds to a global average pooling 2D layer before reaching the concatenation layer. Clinical data is passed to a feature set layer before reaching the concatenation layer. After this concatenation layer, the data is passed through three dense layers, separated by dropout layers. Finally, binary classification is performed to diagnose the patient as either COVID-19 positive or healthy. The proposed model achieves an accuracy of 97% after training on a collection of 270 records from King Fahad University Hospital.

Research in [55] proposed two models—the first uses a DNN on the fractal features of images and the second is a CNN which uses CXR images. The whole model architecture is as follows: a COVID-19 and non-COVID-19 image are fed through feature extraction before being passed to the DNN for classification. The non-COVID input image is also passed to the CNN before feature extraction, and the CNN classifies the image. The COVID input image is passed to the CNN for segmentation, and ground truth is generated. The DNN concluded with an accuracy of 84.11%, which is significantly lower than the accuracy of the CNN, which concluded at 94.6%. CNNs are a type of deep learning neural network suited for image processing. This is due to the use of kernels, which process groups of pixels during each convolution. This allows CNN to capture spatial information, which is extremely useful in the localization of infections. DNNs mostly lack the ability to capture this spatial information due to their inherent architecture thereby leading to CNNs having higher accuracy. The dataset used consists of 812 total images, where 437 are of COVID-19 positive patients and 383 are of healthy patients. In [56], CXR were used to detect infected patients. The authors proposed a hybrid model after comparing 11 different deep learning architectures. The proposed model was optimized using k-fold cross-validations alongside a spotted hyena optimizer for better feature extraction. It was observed that a combination of Resnet-101, with the J48 algorithm decision tree, and the hyena optimizer showed the highest performance across accuracies. The dataset however was smaller with 50 normal and 50 COVID-19 patients. This study [57] developed a deep learning model called XR-CAPS that incorporates a UNet model for image segmentation and a capsule network for feature extraction for the prediction of COVID-19 from CXR images. The dataset consisted of 896 patients who were either healthy, had pneumonia or were COVID-19 positive. The results showed an accuracy of 93.2%, a sensitivity of 94% and a specificity of 97.1%, which outperformed similar models such as ResNet50, DenseNet21, and DenseCapsNet.

In [41], the authors created a system called COVID-CheXNet which used CXR to detect COVID-19. Prior to using ResNet-34 to classify scans as having COVID-19, the authors used contrast enhancement on the CXR and noise reduction technique by applying adaptive histogram equalization and Butterworth bandpass filter. The system achieved an accuracy rate of 99.99%, a sensitivity of 99.98%, and a specificity of 100% using the weighted sum rule at the score level. The model was trained using 400 COVID-19 and 400 normal CXRs gathered from five different sources. Discussions in [58] proposes two experiments to predict COVID-19 in CXR images. The first experiment performs a binary classification between COVID-19 and normal patients. In the second experiment, the classes COVID-19, normal, and pneumonia were considered. The proposed Deep CP-CXR model includes batch normalization first. This is followed by six blocks, which consist of a convolutional 2D layer and a leaky ReLU layer. After this is a max pooling 2D layer, a dropout layer, a flatten layer, a dense layer, a dropout layer, and a final dense classification layer. The accuracy of binary classification was 100%, and the accuracy of multi-class classification was 98.57%. The dataset used 1626 COVID-19 positive images, and 1802 normal images.

Research in [59], the neural network architecture CapsNet was used to classify COVID-19 in CXR images. In contrast to a normal CNN, CapsNet uses capsule layers. High-level capsules reuse the output of multiple lower-level capsules to produce an output vector that measures the probability of an entity existing in the image. This allows a capsule architecture to better capture spatial information, which traditional CNNs degrade through the pooling technique. The model classified images as either normal, COVID-19, and pneumonia. A train/validation/testing split of 60/20/20 percent, respectively, was used on the dataset of 454 COVID-19 images, 4273 pneumonia images, and 1583 normal images. The method of convolutional and capsule neural networks was compared in experiments, in which the proposed capsule model outperformed the conventional CNN architecture. The accuracy of normal, pneumonia and COVID-19 was 77.9%, 88.1%, and 81.2% for the CNN and 86.6%, 89%, and 94% for the proposed capsule network. This study [60] also proposed the use of CNNs for the purpose of classifying CXR images as either healthy, COVID-19, or other viral-induced pneumonia. The dataset used was created from a previous study, which worked to combine public datasets of COVID-19 and other pneumonia images, along with normal images. The final dataset included 10,192 normal images, 3616 COVID-19 images, and 1345 pneumonia images. Multiple CNN architectures were tested—namely ResNet18, GoogLeNet, AlexNet, VGG16, DenseNet161, along with a minimal CNN model created by this study. Each of the popular architectures contained a minimum of 6,000,000 parameters, while the proposed model only contained approximately 130,000 parameters. Every model tested, including this minimal CNN, achieved an accuracy of 97% or higher on the given data. Of these models, ResNet18 held the highest experimental performance by correctly classifying 256 of the total 269 testing images. This ResNet18 model was passed to an end-user application, which was given to a total of 10 experts, each with 12 to 23 years of professional experience in radiological analysis. The experts independently selected 150 images from their own datasets, for a total of 1500 total images being tested. The model correctly classified a total of 1486 of these images, which was confirmed by the experts. The greatest source of incorrect classification came from uncertain images (such as early stages of infection), and in these cases, the experts reported that it was also hard for humans to classify.

The study in [61] similarly proposed a CNN for the use of classifying CXR images as either COVID-19, pneumonia, or normal. The architecture used was based on the popular architecture EfficientNet B5, which had been pretrained using the noisy student approach. Transfer learning was applied to the model in order to fit it to the given classes, and the rest of the deep learning model was based on a prior study. The datasets used by this study were $COVID_{x}$, $COVID_{BIMCV}$, and $COVID_{private}$ which contain 13,958, 10,953, and 305 images, respectively. An equal amount of testing images for each class were selected, with 300 from $COVID_{BIMCV}$ and 150 from $COVID_{private}$. This was the same for validation, with 300 images each from $COVID_{x}$ and $COVID_{BIMCV}$, and 90 from $COVID_{private}$. Five separate random divisions of training and validation datasets were created, each with a respective EfficientNet model. This created five separately trained EfficientNet models, which were combined into an ensemble approach for classification. The final ensemble model was compared to the skills of six radiologists, with 10 months to 15 years of experience. The deep learning model in this study scored higher than all six radiologists on the classification of COVID-19 and showed results comparable to the other models or radiologists for non-COVID-19 pneumonia and normal images. Researchers in [62] propose a model which combines feature maps from trained models to classify images as either COVID-19, viral pneumonia, or normal. Two public CXR datasets from the Kaggle website were used, each with the classes normal, COVID-19, and viral pneumonia. In terms of COVID-19, viral pneumonia, and normal images, the first dataset had 137, 90, and 90 images, respectively, while the second has 3616, 1345, and 10,192 images. The proposed approach involves the use of feature maps from MobileNetV2, EfficientNet0, and DarkNet53 in that order. After these feature maps are concatenated, the NCA optimization algorithm is applied to select the best 82 features out of the 3000 which is given by the concatenated feature maps, which serves to reduce time and cost in the proposed approach. Multiple classification techniques were tested, with SVM classification providing the best results across each of the datasets. The proposed techniques performed well with an average accuracy of 99.05% on the first dataset and 97.1% on the second dataset.

The study in [63] proposed a fusion-based CNN named COVDC-Net to classify CXR images into either three or four classes. The paper claims to address not only the need for rapid and accurate detection of COVID-19 but also the possibility that the currently overworked radiologists might have an increased rate of error due to exhaustion. Two datasets were used in this study—the first contains 4883 total images of the classes COVID-19, normal, and pneumonia. The second dataset explores the use of four classes for the classification task, with 305 COVID-19, 375 healthy, 379 bacterial pneumonia, and 355 viral pneumonia images. The proposed fusion method combines VGG16 and MobileNetV2, both of which are pretrained on the ImageNet dataset. These models were selected as a pair after testing many combinations between VGG16, VGG19, MobileNetV2, ResNet50, and DenseNet. Using confidence fusion, the types of features can be taken from both CNN architectures to create a more effective classification system. An overall accuracy of 96.48% is achieved for the first database of three classes, and an overall accuracy of 90.22% was achieved for the second database of four classes. A note is made to the vague or incomplete representation of metrics from other studies, which may cause repercussions if the techniques are later used in practical settings. The full metrics of this study are displayed. This study [64] used a lightweight CNN with edge computing to efficiently detect COVID-19 in CXR images. To increase the number of samples available in the dataset, DCGAN was implemented. DCGAN is a deep convolutional generative adversarial network, which uses deep learning in an unsupervised approach that replicates the distribution of data in each dataset. This generates new training images, which improved the accuracy of the classification model. For classification, the models MobileNetV2, ResNet18, and VGG19 were tested, and each model achieved functionally high accuracies. In order, MobileNetV2, ResNet18, and VGG19 achieved an F1 score of 82, 87, and 83 percent. The authors conclude that the accuracy achieved through MobileNetV2 is very comparable to the other models while utilizing a much smaller model. The use of edge computing further increases the efficiency of the system by processing information locally instead of relying on a centralized server.

The research done in [65] utilized image processing and a proposed CNN architecture to accurately classify CXR images as COVID-19, pneumonia, or normal. The dataset used in this study is a combination of three separate datasets. A total of 864 COVID-19, 6235 pneumonia, and 4254 normal images were available across these three datasets. These collected images were subjected to preprocessing, which extracted the lungs from the image as the ROI. This ROI is then combined with the original image and given to CNN for classification. The proposed CNN architecture C-COVIDNet is designed to be extremely lightweight and only consists of eight total layers. Five convolutional layers are used, followed by three dense layers, and between each layer, a dropout is used. The last dense layer uses SoftMax activation to generate the class prediction. In total, 97.5% accuracy was achieved using the preprocessing techniques and C-COVIDNet, which beats other state-of-the-art CNNs. The study in [66] proposes a novel model to detect COVID-19 and pneumonia using CXR images in real-time. The dataset used in the study contains 1102 COVID-19, 1341 normal, and 1345 viral pneumonia images. The proposed model COVID-CXDNetV2 uses an implementation of YOLOv2, which is then modified using ResNet. There are 14 convolutional blocks, five max-pooling layers, and four residual connections included in the model followed by a convolution layer, flatten layer and linear layer in that order. In differentiating between the three classes (normal, COVID-19, and pneumonia), the overall accuracy of the proposed model is 97.9%, which outperforms some state-of-the-art models. The study in [67] proposed both an LSTM and CNN model to detect COVID-19 and influenza in CXR images. For this study, a time-series dataset was gathered from Kaggle, UCI repository, and hospitals in Islamabad. For this experiment, 70% of the images were used to train the models, and 30% of the images were used to test. The proposed system is as follows: A series of CT scans are fed into the CNN, which identifies the images as either healthy or infected. If a patient is determined to be healthy, then this information is passed on to medical records. If the patient is determined to be infected, the images will be sent to the LSTM model which classifies the infection as either COVID-19 or Influenza, which would then be passed to medical records. This architecture is designed to better differentiate between COVID-19 and Influenza in the early stages of infection, which poses a challenge due to the similarity between the two infections. Of the two models developed, the LSTM performed the best—the CNN achieved 94% accuracy, while the LSTM achieved 98% accuracy. Both accuracies are high, which shows that the proposed deep learning techniques can accurately differentiate between healthy, Influenza, and COVID-19 using CXR images.

Researchers in [68] explore the potential of semi-supervised neural networks in detecting COVID-19 in CXR when only a small number of training images is available. The dataset contains five image classes—normal, bacterial pneumonia, COVID-19, lung opacity, and viral pneumonia. There are 404 training images for each of the classes, as well as 207 testing images for each class which were collected from a different source than the training images. These images were resized to 224 × 244, and a percentage of data was assigned to be unlabeled data to employ semi-supervised learning. The labeled proportions tested were 80, 60, 40, 20, 10, and 5 percent. These tests also included varied distributions of the majority/minority classes, all to test the robustness of FixMatch. The algorithm used in this study is the FixMatch algorithm, which utilizes consistency regularization and pseudo-labeling techniques. An artificial label is produced for an image with weak augmentation before the model is trained, and the model learns to predict the label with a stronger augmentation of the same image. The FixMatch algorithm utilizes ResNet18 as the underlying model. The proposed methods achieved an accuracy of approximately 78% when the proportion of labeled data was between 80 and 20 percent. In comparison to other CNNs and an ensemble approach, it was on par with the standard supervised learning techniques. The ensemble approach achieved the highest accuracy of 81%, but the proposed method is very effective with data and computational efficiency considered as FixMatch only uses ResNet18. This data also suggests that FixMatch could be robust to data imbalances. Table 2 presents a summary of the deep learning models that were applied for COVID-19 diagnosis using CXR.

A deep learning model to diagnose COVID-19 through the identification of structural abnormalities in CXR images was proposed in [69]. The model architecture uses two convolutional 2D layers followed by a max pool 2d and a dropout layer. Then, three blocks which consist of three convolutional 2D layers, a max pool 2D, and a dropout are used. The model then uses a flattened layer, a dense layer, a dropout layer, and a final dense classification layer. The final accuracy of the model is 97.67%. After data augmentation, the CXR dataset contains 1000 COVID-19 positive images and 1000 normal images. Another deep learning model proposing risk stratification was implemented in [70] using a patient’s CXR images. A modified commercial deep learning model, M-qXR, was used. The model performed well, with 98% of the total interpretations successfully compared to the ground truth. A dataset containing 2.5 million CXR images was used, which contains many pulmonary pathologies such as pneumonia and tuberculosis. The study in [71] provides evidence that CNNs along with transfer learning can be used to train models which can detect COVID-19 in CXR with high accuracy. A dataset of 300 normal images and 200 COVID-19 images was used, where 20 images of each class were set aside for validation, and 50 images of each class were set aside for testing. Data augmentation techniques were used to prevent overfitting and increase variation in the dataset, including a random flip and rotation of 5 degrees (both clockwise and counterclockwise). The best performance of a model used a CNN along with the VGG16 architecture using the transfer learning technique. Not only was the accuracy of this model the highest at 99.3%, but it also had the lowest loss at 0.03 and took the least time to train. This proposed architecture took only two minutes to achieve these results, while the second-best accuracy of 96.6% took almost ten minutes. The proposed combination of techniques provides a faster, more accurate, and adaptable network to detect COVID-19 when compared to the current testing standards.

Data used in deep learning goes through preprocessing, which ensures that the data can fit the input size requirement for the model. It is a common practice to include other preprocessing steps to clean the data, such as the removal of bad samples. A bad sample of an X-ray used in COVID detection could have labels written over the data or be extremely noisy. Preprocessing can also include steps that increase the size and quality of the dataset. Image augmentation uses one image to create multiple versions which slightly deviate from the original by methods such as zoom, rotation, and cropping. This prepares the model for these types of deviations and increases the performance on diverse data. There are additional techniques that actually generate new images to match existing data, often with the goal of balancing a mismatched dataset. Synthetic Minority Over-Sampling Technique, or SMOTE, is used to increase the number of images or balance data pertaining to different classes. These techniques are often used when a large dataset is not available for training, which was an extremely common setback at the beginning of the COVID-19 pandemic as there was little data collected, and it was often private. If CXR images are insufficient for the deep learning task, more data can be included in the form of clinical variables. With the increase in relevant data, deep learning models can give more accurate predictions.

The study in [72] proposed novel architecture using convolutional and temporal neural networks to detect COVID-19 in CXR. For the study, some datasets were combined for a total of 1670 COVID-19 images, 1672 normal images, and 1670 pneumonia images. These images were preprocessed using the empirical wavelet transform, which is useful for removing noise and increasing the resolution of the CXR images. Four models were tested alongside the proposed model, including InceptionV3, ResNet50, and ResNet50-TCN. In the proposed RESCOVIDTCNnet, features are extracted from these images using ResNet50, which are then passed to TCN. ResNet50-TCN is the proposed architecture, the difference being that RESCOVIDTCNnet includes the Empirical Wavelet Transform (EWT) preprocessing. Classification is then performed by either SVM or MLP, with MLP performing the best. In both cases, RESCOVIDTCNnet achieved an accuracy of over 99%. The paper reports that the proposed model outperforms the other models with a correlation coefficient of 0.999, and a standard deviation of 0.1.

### 4.2. Diagnosis Using Primarily CT Imaging

In [73], the authors talk about generic background information and human processes for a diagnosis via deep learning. Other models are brought up with their accuracies, with an emphasis on the models the research team recreated themselves. Dataset used had 349 COVID-19 images from 216 patients and 463 from non-COVID-19 patients. The researchers found VGG-19 worked best in their studies, at 94.52% accuracy. Research in [74] creates an argument for utilizing the VGG19 model while comparing Alexnet, Googlenet, Inception v3, VGG16&19, Shufflenet, Mobilenet v2, Resnet 18, 50 & 101. VGG19 showed the best performance across “covid”, “normal” and “pneumonia” case datasets. COVID-19 datasets used were large, open-source verified sets.

Authors in [75] used both chest X-rays (CXR) and CT scans to propose a CNN model named CoroDet. This model is able to perform a 2-class, 3-class and 4-class classification using a 22-layer sequential CNN which consists of convolutional, max pooling, dense layer, flatten layer, and three activation functions, namely Sigmoid, ReLU, and Leaky ReLU. Training on a dataset called COVID-R containing 7390 total CXR, 2843 COVID-19, 3108 Normal, and 1439 Pneumonia (Viral + Bacteria) the researchers ended with an average accuracy of 91.2%. Research in [76] used CT and CXR images to evaluate pre-trained deep neural networks in their ability to detect COVID-19. Data augmentation is used to increase the size of the training dataset to avoid overfitting and increase model performance. Among the tested models—ResNet50, InceptionV3, VGGNet-19, and Xception—it was found that VGGNet-19 performs the best on the CT dataset with 88.5% precision, 86% recall, 86.5% F1-score, and 87% accuracy. Xception performed the best on the CXR dataset with precision, recall, F1-score, and accuracy equalling 98%. Notably, the VGG-19 model performs the best across the two datasets with an average of 90.5% accuracy. The dataset used contained 3000 healthy scans and 623 COVID-19 positive CT scans.

Eight CNN CovNet models were compared in [77] in their ability to diagnose COVID-19 in CT scans as well as CXR images. A ninth model was proposed, which consists of four blocks of a 2D convolutional layer followed by a dropout layer, an activation layer, and a max pooling 2D layer. After these four blocks, there is a flattened layer, a dense layer, a dropout layer, and finally a dense classification layer. This proposed model achieved the highest accuracy at 98%, a precision and F1-score of 96%, and a recall of 95%. Their CXR dataset contains a total of 4107 COVID-19 positive images and more than 4105 healthy images. Their CT dataset contains a total of 1605 COVID-19 positive images and 1626 healthy images. Hybrid models were also implemented with success. Research in this domain was targeted toward rapid real-time COVID-19 detection. In [78], the authors highlight how computed tomography (CT) or CXR have become a significant tool for quick diagnoses. Thus, it is essential to develop an online and real-time computer-aided diagnosis (CAD) approach to support physicians and avoid further spreading of the disease. Using the ResNet50 model it was able to achieve a 98% accuracy with the CXR dataset. This paper highlights how CXR can get great accuracy and be a lot easier to get from patients. CT images can effectively complement reverse transcription-polymerase chain reaction testing. Using the COVIDx-CT dataset containing 104,009 CT images from 1489 patients and putting them through the COVID-Net CT-2 and ResNet-v2 models, research in [79] was able to achieve an accuracy of 98.7% which is comparable to CXR showing how CT and CXR can get the same accuracy.

In [80], a self-named DREnet model comprised of a pretrained ResNet 50 model using a Feature Pyramid Network to extract top-K details, before being passed into a multi-layer perceptron to make predictions. The goal of the model was the accurate diagnosis of COVID-19 in patients. The model was run on a dataset of 88 COVID-19 patients, 100 patient cases of bacterial pneumonia, and 86 healthy people. In [81], used a ResNet(50) model, with a proposed accuracy of 98.8%, with a dataset of 2267 CT scans, including 1357 COVID-19 patients and 1235 CT scans from non-COVID-19 patients. The paper confirmed that no image augmentation is necessary for diagnosing COVID-19 and it can be done using highly compressed JPEG files as well. Pre-trained models of ResNet 18,50, and 101 alongside SqueezeNet were used in [82] to detect COVID-19 from CT scans. Using the familiar 349 COVID-19 images from 216 patients, with 463 from non-COVID-19 patients, their findings showed that ResNet18 worked best for their dataset, with a validation accuracy of 97.32%, training 99.82%, and testing 99.4%.

In [39], the authors used 48,260 CT scan images from 282 normal patients and 15,589 images from 95 patients with COVID-19 infections. The researchers used an image pre-processing and filtering algorithm to remove CT image slices that did not represent the region of interest associated with COVID-19 detection. Much of this filtering approach was based on removing CT slices from the top and the bottom of the scan, and also implementing a threshold for the Hounsfield Units that is suitable for lung tissue analysis. Using this scheme, lead to higher accuracy and less false detection. When ResNet50V2 model was used on the modified dataset, the researchers were able to achieve an accuracy of 98.49% using a single image classification stage. In [83], the authors created an effective deep learning algorithm to screen for COVID-19. The dataset included 1065 images from 259 patients. As a pre-processing stage, the authors delineated tissue in CT images using Region of Interest (ROI) for the training and validation step. This patch-based method allowed the model to focus on features that are evident during the early stages of COVID-19 prognosis compared to viral pneumonia. Proposing an Inception-v3 model with these patch-based ROI, they achieved an accuracy range of 79.3–89.5% through their tests.

In [84], the authors compared pre-trained models to detect COVID-19 in CT scan images, with the attempt to offer a model to perform autonomous prediction. Among the models tested are InceptionV3, ResNet50, VGG16, and VGG19. ResNet performed the best, with an accuracy of 98.86%. The research used a CT dataset of 7395 COVID-19 positive images and 6018 suspicious case images, while most models proposed earlier require a lot of computing resources, researchers in [85] made an effort to diagnose COVID-19 with limited resources. With the use of a fine-tuned MobileNet-V2 model being fit on a real-world dataset consisting of 2482 CT scan images, the authors were able to achieve an accuracy of 85.6%. The proposed model was significantly smaller at around 8.45 MB compared to the similarly sized VGG16 model at 512 MB, DenseNet 201 at 70.9 MB and VGG19 at 532 MB. The average classification time on test data images was also less at 21.3 s with 43 msec/image compared to 315 s for VGG-19. In [86], the authors proposed COVID-AL, a weakly-supervised COVID-19 detection module using a pre-trained UNet model for transfer learning. Their model is described as a “tailor-designed 2D U-Net”, cropping images to 352 × 320 before input. The significance of this research was that the authors were able to train and test the model on separate datasets. The model began by segmenting the lung region from 402 CT scans using the Non-Small Cell Lung Cancer (NSCLC) dataset. This objective of this stage is similar to [83], however, instead of extracting ROIs, the UNet was creating a segmented mask of the lung from an entire CT scan which could have several slices. The UNet segmentation is then used on 617,775 CT scan slices from 4154 patients in the China Consortium of Chest CT Image Investigation (CC-CCII) dataset. Through their model, they attained an accuracy of between 96.2 and 96.8%.

Research in [87], examines the problem with deep learning models requiring labeled training datasets. It takes pre-trained models ResNet18, ResNet50, and ResNet101 and uses them for the automatic detection of COVID-19. ResNet50 performed the best with a recall of 98.8% and an F-1 score of 98.41%. Different layers of the model are explored through support vector machine, logistic regression and K-nearest neighbor classifiers. A fusion of the classifiers was deployed as a strategy and the recall score increased to 99.2% while the F-1 score increased to 99.4%. Research in [88], examines the challenge of diagnosing patients with COVID-19 and differentiating it from pulmonary edema. A machine learning model EDECOVID-net was used to differentiate using lung Computed Tomography radiomic features. The model was able to distinguish between COVID-19 and pulmonary edema with an accuracy of 0.98. Research in [89], examines early diagnosis of COVID-19 through a deep learning model approach combined with CT image analysis. The model is based on convolutional long short term memory and was tested on CXR images. The performance of the model had an accuracy of 97.3% at 50 epochs. This [90] paper proposes using a CNN model for the diagnosis of COVID-19 patients using CT images. This predictive model was used for binary classification of other pulmonary diseases. The binary classification had an accuracy of 98.79%, a precision of 94.98%, and a sensitivity of 98.85%.

The researchers in [91] address the need for quick and automated detection of COVID-19 by proposing a transfer-learning deep CNN to classify images as normal, COVID-19, and pneumonia. The transfer learning is performed using pre-trained DensNet121, in which the model is given images augmented with random flips, rotation up to 10 degrees, zoom up to 5%, and changes to brightness and contrast. The model weights are adjusted to fit the new inputs and outputs. Data preprocessing occurs before images are passed to the model during testing, which includes a resize to 224 × 224 × 3 and normalization. DensNet121 is pretrained on the popular ImageNet dataset, which contains over 1.2 million images belonging to 1000 classes. A publicly available dataset was then used for transfer learning, consisting of 180 images per class. In total, 80% of images were used to train, and 20% of images were used to validate the model. In experiments, the proposed model showed an accuracy of 96.52% across the three classes. Compared to other popular architectures, this was the highest performing model with at least 2% higher accuracy than other models.

Research in [92], utilizes preprocessing techniques and a bidirectional LSTM to classify images as normal, viral pneumonia, lung opacity, or COVID-19. The image dataset used contains 3616 COVID-19 images, 10,192 normal images, 6012 Lung opacity images, and 1345 viral pneumonia images. The first step of preprocessing was to use modified histogram equalization to enhance contrast. The equation was modified to change the level of the histogram based on the contrast level of the input image. Preprocessing also included an extended dual-tree complex wavelet transform with trigonometric transform to extract effective features. A custom optimization technique was used to avoid overfitting by selecting features with reduced dimensionality. These preprocessing techniques increased the final accuracy of the model from 97.34% to 99.07%, proving their effectiveness. Classification was performed by the adaptive dual-stage horse herd BiLSTM architecture. This model uses an embedding layer, followed by a bidirectional LSTM (forward and back), then a concatenation layer. This concatenation layer leads to another bidirectional LSTM and concatenation layer, which is passed to an optimized attention layer, which gives an output class. When compared to ResNet50 and AlexNet models, the proposed model obtained the highest accuracy at 99.07%. The accuracy per class was always greater than other models by at least 1.73%.

The study in [93] proposed a data-efficient deep neural network to account for the lack of data available during a pandemic situation. Two datasets were employed, and a general adversarial network was utilized to increase the number of samples while creating different types of lung deformations. This novel approach was added to the methods because while data augmentation increases accuracy in varied datasets, it is ultimately using data with the same exact deformation in the augmented images. The first dataset—COVID-19-CT—has 324 train and 40 test COVID-19 images, and for normal images, there are 293 training sets and 37 test sets. The second dataset—MosMed—has 168 train and 20 test images for both COVID-19 and normal images. After augmentation and GAN implementation, the COVID-19-CT dataset contains 2172 train and 37 test normal images, along with 1893 train and 37 test COVID-19 images. The MosMed dataset ends up with approximately 1200 images for both COVID-19 and normal. The highest AUC value of 0.89 was achieved using ResNet18 on the first dataset along with augmentation and GAN. Without the GAN implementation, the AUC value is only 0.77, which proves the utility of GAN for synthetic image generation. The study in [94] proposed a deep learning model to segment COVID-19 infection in lung CT scans using the UNet architecture. For this purpose, a dataset of 3138 lung CT images belonging to 20 patient scans was used, each with a respective label and segmentation verified by expert radiologists. A training/validation split of 70/30 was used to verify the accuracy of the model during training. The proposed diagnosis system consists of four separate phases—lung segmentation, infection segmentation, COVID-19 classification, and 3D reconstruction. The lung segmentation phase utilizes contrast-limited adaptive histogram equalization (CLAHE) as a preprocessing step and cropped away any background slices. The U-Net architecture is then used to segment the lungs. The infection segmentation phase performs the same preprocessing of CLAHE and background slice removal, segmenting only the infection. The COVID-19 classification phase uses a three-layer CNN with fully connected layers and softmax activation to then classify the infection segmentation as either COVID-19 or normal. The final phase gave a 3D reconstruction which was used to identify the rate of infection, quantified by the ratio of the infected lung to the healthy lung. For the final classification, an accuracy of 98% was achieved. By using these techniques, the model was able to achieve high performance with a limited training dataset.

The study in [95] utilizes lung segmentation to increase the accuracy of a CNN in the classification of COVID-19 in CT images. A Kaggle dataset is used, which contains a total of 1834 images along with masks. To segment the lungs, the techniques of canny edge detection, thresholding, and UNet are compared. The Intersect over Union scores indicate that UNet greatly outperforms the other techniques. A CNN is then used to classify the lung as either COVID-19 positive or negative. The final accuracy achieved through these techniques is 95%. The inclusion of segmentation both increased the accuracy and decreased the loss, proving the utility of the technique. The study in [96] proposes a novel CNN model named DCML to detect COVID-19 in CT and CXR images. A binary classification occurs, labeling images as either COVID-19 positive or negative. For this research, three total datasets were used to train and test the models. Each dataset provided separate characteristics—the first dataset had less than 600 total images to train on, the second dataset was overall fair, and the third dataset had a large data imbalance. In total, 70% of each dataset was used for training, while 30% was reserved for testing. Each training dataset is augmented using the Fast AutoAugment method. Deep mutual learning is then employed, and the models are strongly fused. Contrastive learning is included in the knowledge transfer during the mutual learning phase, which increases the model’s performance in distinguishing between classes. The final proposition of this research is the adaptive model fusion strategy, which utilizes the correlation between the heterogeneous networks to better mimic the learning and decision-making processes of trained radiologists. The feature maps from two separate networks are combined using a feature fusion model, which creates a new fused feature map. This is then passed to the fully connected layers and a SoftMax classification. Across each of the datasets, the proposed DCML using Fast AutoAugment provided the highest accuracy when compared to a variety of fine-tuned popular CNN architectures.

The study in [97] proposed the novel DMDF-Net for the segmentation of COVID-19-related lesions in CT scans. For this deep learning model, two datasets were used—the first contains 2049 images with a COVID-19 infection ground truth, and the second contains 3520 images with masks for the left and right lungs along with the COVID-19 infection mask. Before images are passed to the model, a preprocessing step occurs where the color and contrast are adjusted. The DMDF-Net model uses dual multiscale feature fusion in both the encoding and decoding phases, which increases the accuracy of the segmentation. The encoding phase is based from MobileNetV2. Postprocessing is also implemented, in which the lung mask ROI is applied to the original CT image for further processing the data and segmenting the infected regions. This technique was observed to increase the accuracy of the model. The final model—including all pre- and post-processing techniques—was able to outperform other state-of-the-art models with a DICE score of 75.7% and an IOU of 67.22%.

Research in [98] aimed to contribute a decision-aiding tool that can detect COVID-19 in CT images. The images are preprocessed first by a random window level between −500 and −600 HU, and a window width of 1200 HU. The randomness of the window level is chosen to account for the discrepancies between CT scans taken with different machines. The second step in preprocessing includes a normalization to between 0 and 255, and the removal of unrelated features or noise in the image. The model architecture is an end-to-end segmentation model based on the UNet architecture, which utilizes an encoder and a decoder along with the SoftMax activation to generate a predicted mask. To tackle the imbalanced class problem where the infection is always between 0 and 20 percent of the image, a custom loss function is used. The loss is a class-balanced cross-loss function, which was observed to effectively handle the imbalance. The DICE score of the proposed model, including preprocessing, was 83.3 and 83.4 percent for datasets one and two, respectively. Compared to the performance of popular baseline architectures, this model achieved the highest DICE score. The study in [99] fine-tuned existing models VGG16 and miniVGGNet for the prediction of COVID-19 on CT imaging. In total, 6368 CT images were used from the Corona Hack-Chest X-Ray dataset of which 3184 were COVID-19 positive and 3184 were normal. The efficiency of each model was evaluated with VGG16 having a precision of 0.98, recall of 0.81 and F1 score of 0.89 for COVID-19 while miniVGGNet had a precision of 0.97, recall of 0.89 and F1 score of 0.93 for COVID-19. Both models performed best with data augmentation with VGG16 achieving an accuracy of 89% while miniVGGNet had an accuracy of 93%.

In [100], the authors proposed using a deep learning model for quantitative assessment of COVID-19 CT imaging. A total of 14 patients were reviewed with both initial and follow-up CT. The deep learning model was based on the CNN VB-Net, which was used for automatic segmentation and delineation of affected regions. The results showed the percentage of infection was 3.4% for the entire lung, with a percentage of infection decrease in the follow-up period for all 14 patients. The study in [101] developed a model called Deep Covix-net that used image segmentation along with a machine learning algorithm to accurately predict and diagnose COVID-19. The dataset that was utilized in this study was an open-source Kaggle and GitHub repository that contained 9000 CXR images, of which 3000 are COVID-19 positive, 3000 are normal, and 3000 have pneumonia. It also contains 6000 CT images, of which 3000 are COVID-19 and 3000 are normal. Feature extraction and selection involved using advanced image segmentation techniques such as texture, grey-level co-occurrence matrix (GLCM), grey-level difference method (GLDM), fast Fourier transform (FFT) and discrete wavelet transform (DWT). The statistical features were combined into the Deep Covix-net model and a random forest classifier for classification. The CXR dataset had an accuracy of 96.8% while the chest CT images had an accuracy of 97%. Table 3 presents a summary of the deep learning models that were applied for COVID-19 diagnosis using CT.

### 4.3. Diagnosis Using Ensemble Techniques

An ensemble approach to combining results from multiple deep learning models was also reported in [102]. Here authors used VGG-16, ResNet50, and Xception models and then combined their results to generate a final prediction on 2482 CT-scan images, 1252 of which are scans with positive COVID-19 cases. With a 98.79% accuracy, these models have proven themselves very useful. Another stacked ensemble approach was explained in [103] which looked at the predictions of VGG19, ResNet101, DenseNet169, and WideResNet 50-2. Five datasets were used to verify the proposed approach with 15,286 CT scans and 3120 CXRs. The proposed model was compared with 11 existing models from previous research. The proposed model outperformed in all five datasets with accuracy values of 85.45%, 93%, 99%, 99.75% and 91.5%.

Research in [104] combines the use of Stacking and Weighted Average Ensemble (WAE) on popular models (VGG19, ResNet50, and DenseNet201) in an ensemble learning approach to diagnosing COVID-19. The input is passed to the fine-tuned popular models mentioned before, and the output is passed to Random Forest and Extra Trees classifiers. That output is passed to a Logistic Regression level, which then produces an output of either positive or negative. The F1 scores of the WAE model were the highest, at 98.65% and 94.93% respectively for the two CT datasets. Two separate CT scan datasets were employed—the first contains 1252 COVID-19 positive images and 1230 healthy images. The second contains 349 COVID-19 positive images and 463 healthy images. Researchers in [105] made an effort to create a generalized detection model that can look at unknown datasets and detect COVID-19 with significant accuracy. A meta-classifier-based approach along with EfficentNet-based pretrained model was used to extract significant features from datasets. These features were then reduced using Principal Component Analysis (PCA). The final classification layer was a 2-stage approach. In the first stage, Random Forest and Support Vector machines were used to bag the prediction results followed by a logistic regression classifier to delineate the final outcome of COVID-19 or not. The authors tested the model on 8055 CT images and 9544 CXR. Seven variants of EfficentNet were trained based on image scaling along with 18 variants of deep learning models used by previous researchers. The proposed approach reported a precision and recall of 0.99 for COVID-19 images and 1 for Non-COVID-19 images.

The excerpt from [106] used a binary classification IKONOS-CT to differentiate COVID-19 patients from non-COVID-19 using CT images. The classifiers that were used were multilayer perceptron, SVM, random tree, random forest, and Bayesian networks. The best feature extractor was Haralick and SVM which had an accuracy of 96.994% and recall of 0.952. Research in [107], uses machine learning techniques to identify COVID-19 through extracted feature fusion. Curvelet Transform, Gabor Wavelet Transform and Local Gradient Increasing Pattern were used for fusion and then used to classify CXR images. The machine learning classifiers used were Discriminant Analysis, Ensemble, Random Forest, and SVM. The model had an overall accuracy of 96.18%, a precision of 95.46%, a sensitivity of 96.98% and an F-1 score of 96.21% with SVM. Research in [108], classifies the severity of COVID-19 positive CT images through feature extraction. The text features are then classified using Random Forest. The images are then placed into four levels of severity of COVID-19, which had an accuracy of 90.95%.

Research in [109], develops a machine learning-assisted model for the detection of severe COVID-19. They use blood tests clinical variables along with quantitative CT parameters such as volume, percentage, and ground glass opacities. Correlation between the variables was first done using a Pearson test, then features were selected through an independent t-test and least absolute shrinkage and selection operator regression. Support vector machines, Gaussian Naïve Bayes, K-nearest neighbor, decision tree, random forest and multi-layer perceptron algorithms were all used as classifiers. For the selected features, lesion percentage contributed the most to classification. Random forest had the highest accuracy for the identification of severe COVID-19 cases at 91.38%. Research in [110] used a hybrid of deep transfer learning and machine learning to classify CT scans as either COVID-19 or normal. For this study, a dataset of 1252 COVID-19 images and 1230 normal images was collected from a hospital in Brazil. The approach uses a popular CNN architecture to extract the hidden features of the images, which are then passed to one of two machine learning algorithms—support vector machines and k-nearest neighbors (KNN). The transfer learning approach was chosen to account for the small dataset, as the model only needs to adapt the existing weights to fit the new data. The final layer of each tested deep learning architecture was a pooling layer, which consisted of the extracted features. This was then passed to the chosen machine learning algorithm, which performed the final classification. A final accuracy of 98.2% was achieved using the features extracted by the ResNet50 architecture and the SVM classification. The ResNet50 architecture also returned the highest accuracy when used with the KNN classification at 97.5%. Both proposed approaches outperformed other studies which utilized the same dataset.

The study in [111] proposes a feature fusion deep learning architecture to extract features, then classify them as either being COVID-19, pneumonia, or normal. A dataset of 6518 CXR images was used for this study, with 20% of this being devoted to testing. As a part of preprocessing, a ResNet34 architecture was used to segment the lungs in the image. In the proposed architecture, the original CXR image (without preprocessing) is passed to DenseNet while the ResNet34 segmented output is sent to VGG16. Both DenseNet and VGG16 extract features and using the ensemble learning technique they are concatenated and passed to the attention layer for classification. This attention layer includes a global then category attention block, which allows the model to overcome data imbalance and obtain more detailed information on smaller lesions. After a global average pool and dense layers, a softmax activation is used to classify the image. Multiple aspects of the model were fine-tuned, which resulted in a testing accuracy of 97.3%, higher than the popular model architectures used for comparison. The second-best model was DenseNet169, which obtained a testing accuracy of 94.7%. The study done in [112] uses deep learning to classify patients as COVID-19 positive or negative from their CT scan images. The Basrah Dataset was used, which has 1181 positive cases and 242 negative images from a total of 50 patients. In total, 818 images were used for training the model, 694 of which being COVID-19 positive. In total, 605 of the images were used to test the model, with 487 of those being COVID-19 positive cases. Images would first go through preprocessing, in which the image was converted to grayscale and resized. Then, feature extraction is performed by the popular VGG16 CNN architecture which ends in a max pooling to decrease the feature map resolution. The image feature map is then passed to the fully connected layers, which use SVM for classification. Through testing the model, a reported 99% F1 score was achieved.

The study in [113] proposed a novel lung segmentation technique and a classification CNN to classify the COVID-19 disease in CT images. The COVID-19 CT dataset was utilized for training, which includes 349 positive and 397 negative CT scans. Image augmentation, such as rotating the image by 90, 180, and 270 degrees was used to increase the amount of data available for the model. Before being given to the model, images are segmented using the thresholding technique to create an initial mask. After this, the GrabCut technique is utilized to ensure that unnecessary information was not removed or included in this mask. For the proposed classification methods, the popular model architectures of a convolutional deep belief network (CDBN), HRNet, and VGGNet are fused in the fully connected layer to create an ensemble learning voting system. This provides efficiency along with accuracy—the proposed model achieved an accuracy of 95%, which outperforms all the state-of-the-art models which used the COVID-19 CT dataset. A deep learning method is proposed by the study in [114], which uses ensemble transfer learning to detect COVID-19 in CXR images. For this, a dataset was used which consists of an equal 1000 images for both positive and negative cases. A split of 60/20/20 percent was used for training, validation, and testing, respectively. Images were resized to fit various popular architectures, which the study compared for transfer learning, and determined that ResNet50, DenseNet201, InceptionV3, VGG16, VGG19, Xception, and MobileNetV2 performed the best. Each of these models was then used in an ensemble approach—after every model gives an output, they are combined in the ensemble learning step which performs a weighted classification based on those inputs. MobileNetV2 was the fastest performing model, but the ensemble approach achieved a precision of 100% with slightly more time to run.

Research in [115] used Light Gradient Boosting machines as a primary model to predict COVID-19 infection, development of Acute Respiratory Distress Syndrome, Intensive Care Unit admission, and risk of mortality. It used clinical variables along with CT images in its analysis. The results showed COVID-19 infection AUC of 0.790, development of ARDS at 0.781, ICU admission at 0.675, and risk of mortality at 0.759. Finally, the authors in [116] explored the role that CT imaging plays in the prognosis of COVID-19 using machine learning algorithms. A mixture of clinical and laboratory features along with CT imaging was used and reduced to seven predictors. Decisions trees, k-nearest neighbors, SVM and ensemble learning were used as classifiers. The results showed SVM with the best classification at 73.7% accuracy and 0.69 AUC score.

### 4.4. Diagnosis Using Additional Data

Research in [117] utilized the approach of lung ultrasound (LUS) images to diagnose COVID-19 in patients. A mixture of a CNN and regularized spatial transformer network (RSTN) were used. Binary and quaternary labeling was used on the videos—a binary label would describe a video as pathological or normal. A quaternary label would describe a video based on the percentage of frames containing signs of pathology. The quaternary labeling made predictions much more efficient than a frame-based predictor without much sacrifice of accuracy. The frame-based and quaternary-based accuracies were 90.18% and 88.73% respectively. The dataset consisted of videos, each contributing to the total number of frames for each category. After the criteria were met, researchers were left with 507 right anterior, 2071 right posterior lower, and 1249 left posterior lower images or frames—each referring to a separate section of the chest that was scanned. Research in [118], uses a deep learning model to diagnose COVID-19-related pneumonia via Ultrasound. The goal was to classify three categories of normal, interstitial abnormalities, and confluent abnormalities. The neural network used a two-stream approach that used red–green-blue channels and velocity information. The results showed that determining interstitial abnormalities had an F-1 score of 0.86. Research in [119] used multiple deep learning architectures on the POCUS dataset which contains 3326 LUS of three classes namely COVID, normal, and pneumonia. Results indicated that InceptionV3 produced the best outcome at about 89.1% accuracy and 97.1% ROC.

Most of the research revolving around CT images was binary classification. Few models were also developed to predict the severity of the disease in patients. In [120], the authors explored the severity of COVID-19 via a CT scan. A novel VB-Net model was created and achieved 91.6% severity detection accuracy in tests across a dataset of 249 COVID-19 images for training, and 300 for testing. In [121], the authors used an AI-severity score to identify the severity of the COVID-19 patient based on age, sex, oxygen saturation, urea, and platelet counts. This multivariate model used logistic regression and cross-validation folds. EfficientNet-B0 pre-trained on the ImageNet public database and ResNet50 pre-trained with MoCo v2 on one million CT scan slices from both Deep Lesion and The Lung Image Database Consortium (LIDC) as a base for their model. Their research identified that elderly people and the male population are at more risk. Clinical variables associated with disease severity included oxygen saturation, respiratory rate, and diastolic pressure. The proposed model outperformed 11 of the previously published similar research such as COVID-GRAM, the NEWS2 score, and the 4C mortality score. The initial training dataset had a total of 1003 patients from Kremlin-Bicêtre hospital and Institut Gustave Roussy hospital. Almost 506,341 CT images were analyzed where each patient had almost 517 slices per scan. The proposed severity model also reported high AUC for 150 left-over patients from these two hospitals. Table 4 summarizes the results from these two sections.

## 5. COVID-19 Prognostic and Longitudinal Models

In this section, we focus on how machine learning was utilized for longitudinal study in COVID-19 diagnosis, while diagnostic studies are relevant for detecting the importance of COVID-19 detection, longitudinal studies are more applicable for real-time monitoring and prediction. However, these studies present unique challenges as data needs to be collected over a longer timeframe. Figure 11 shows the number of papers that were reviewed in this article with significant results for real-world application. Results indicate a stark contrast to the number of papers reviewed for diagnostic purposes which further support the claim above.

### 5.1. Prognostic Models

One of the early research surveys on prognostic models was reported in [122]. The purpose of this paper was to critically review the value of published or preprint reports of prediction models used for diagnosing potential COVID-19 infected patients, the prognosis of patients with COVID-19, detecting the risk of people being infected by COVID-19, and the likelihood of hospital admissions. The significance of research towards the COVID-19 pandemic is to help alleviate the burden on the health care system while providing the best care for patients through accurate diagnosis and information on prognosis. Prediction models using variables and features to estimate or predict such things could be of huge assistance to medical staff and help with the strategic allocation of healthcare resources. However, the review also indicated that several models were poorly reported with optimism, and high risk of bias, and concluded with the prediction models not being recommended for application in the healthcare systems yet. The paper reviewed a total of 51 studies that described 66 prediction models. The 66 models can be divided into three main categories; 47 models were for detecting COVID-19 of which 34 were based on medical imaging. In total, 16 were prognostic models that were used for predicting mortality risk, progression to severe disease and length of hospital stay. The remaining three models were used for predicting hospital admission from pneumonia or other events in the general population. Methods of machine learning, artificial intelligence, score, deep learning, and regression were used in the prediction models. A model’s predictive performance was evaluated using measures used in the publications, but a C-index is used to summarize discrimination (the extent to which predicted risks discriminate between participants with and without the outcome). Calibration intercept is another metric used to quantify calibration (the extent to which predicted risks correspond to observed risks). Nearly half of the predictive models used data on patients with COVID-19 from China. The rest used data on patients from Singapore, internationally or simulated data. As these studies mainly used case-control sampling or an unclear method of data collection, the diagnostic studies might not be representative of the targeted population the models were being developed for. All features used in the data include medical information regarding the person, such as personal information (sex, age), medical history and metrics used for diagnosing COVID-19. When applied, mortality risk was often too low for high-risk patients and too high for low-risk patients, resulting in a low calibration score. The models also had a risk of bias and discrimination with 4 of the 16 having a high C index of 0.9–0.98.

COVID-19 severity assessment (COSA) score was developed in [123] where data was collected from 198 COVID-19 positive patients at the Insel Hospital Group in Bern Switzerland. Demographic data, medical history and lab values were used from the patients to train machine learning models to predict the outcomes of death or ICU admission. They also used the data from 459 patients who tested positive for COVID-19 at the hospital as a validation dataset. Researchers developed the COSA score ranging from 0 to 10, with a higher score meaning a higher probability of a severe outcome with COVID-19. For the lab tests, thresholds of whether the score points would be added were calculated by looking at graphs of the parameters against severity using the LOESS function and using Decision trees. They determined how many score points should be added for a given variable using logistic regression. The score had an AUROC of 0.934 on the training dataset. When the score was used to predict severe outcomes on the validation set it received an AUROC of 0.85. In the study, they also used several machine learning models to predict the outcome of severe hospitalization. Of the machine learning algorithms, support vector machines performed the best in predicting severe outcomes with an AUROC of 0.96. The median AUROC scores over the cross-validation folds for the different machine learning models ranged from 0.86 to 0.96.

In order to predict ICU admissions, the authors in [124] used data from the Mass General Brigham Healthcare Database which included demographic, data, comorbidities, and clinical lab data of COVID-19 patients admitted to emergency departments at Massachusetts General Brigham hospitals. The training dataset consisted of 3597 patients. The study evaluated 18 machine learning algorithms in predicting the negative outcomes of ICU admission within five days and mortality within 28 days and found that ensemble-based models performed the best. Models had mean F1 scores > 0.81, and ICU admission where all models had F1 scores > 0.83. When predicting ICU admission, the Linear Discriminant Analysis, MLPClassifier and Logistic Regression models performed similarly to the ensemble-based models. When predicting mortality, they found GaussianNB and Linear Discriminant Analysis models performed similarly to the ensemble-based models. Finally, when predicting ICU, they also found that the ensemble-based models had lower Brier scores than the other models when predicting ICU admission which indicates more accurate predictions. When tested on a validation data set, the performance of all models dropped but the ensemble methods performed the best overall in predicting ICU and mortality with F1 > 0.4 and PR AUC > 0.5.

A research study in [125] evaluated several machine learning models like the random forest, extreme gradient boosting and neural networks to predict several outcomes including death, ICU admission and the need for mechanical ventilation. This research took place at a Hospital in Sao Paulo, Brazil. The dataset was formed from 1040 COVID-19 positive patients using their demographic and clinical lab data. Each algorithm was trained and tested on the individual outcomes. The authors also investigated a combination outcome to verify the predictive performances of the model. All models had an AUROC over 0.91 and the models not trained on the predicted outcome were within the 95% confidence intervals of the results on the ones that were trained. This is significant in that the model is predicting with high accuracy an outcome it was never trained on. In addition, they found that Random Forest was the best algorithm when training on the ICU and Ventilation Outcomes, as well as when training on the ventilation and death outcomes. Extreme Gradient boosting was the best algorithm when training on ICU and death outcomes.

A prognostic model was developed in [126] that utilized deep CNN with CT scans along with a gradient boosting model while using clinical data to predict the risk of deterioration in patients. The model had an AUC score of 0.758 in predicting deterioration within 96 h for patients. The AI deterioration model used three separate algorithms to generate predictions. It consisted of two deep neural networks based on Globally-Aware Multiple Instance Classifier (GMIC) and the gradient boosting model which takes into account clinical data. One of the neural networks(COVID-GMIC) and the gradient boosting model is used to predict the overall risk of deterioration whereas the other neural network (COVID-GMIC-DRC) predicted how the risk of deterioration will change over time by calculating deterioration risk curves. These models were trained on 3661 patients. The combined model approach consistently performed better than the GMIC baseline over all prediction time windows. The authors also compared their COVID-GMIC model to radiologists at New York University (NYU) Langon Health and found their model to be comparable to the radiologist’s in terms of AUC and PR AUC scores. A simplified version of their model that analyzed chest radiographs was deployed in NYU Langon hospital which processed the images quickly without the use of GPUs. In total, 375 exams were collected at the hospital and the model deployed achieved an AUC of 0.717 and a PR AUC of 0.289.

Research in [127] presented a model that can be used as an early warning system for the detection of COVID-19 progression within a patient. The model had an optimized solution for getting a diagnosis for potential treatment, better increasing the allocation of medical resources, improving emergency response capacity and decreasing mortality rates. The model used clinical data and sequences of CXR scans. The model could automatically identify indicators that contributed to the malignant progression of COVID-19 based on several factors. Other similar models that this model was compared to were based on linear discriminant analysis, support vector machine and multi-layer perceptron. The data was obtained using patients from two hospitals in Wuhan. In total, 1040 patients who had mild COVID-19 pneumonia symptoms upon admission to the hospital were included in this study. In total, 336 (32.3%) of the patients were considered to have a malignant progression that was severe, or critical and the remaining 704 (67.7%) patients were considered noncritical. Data were divided into three different cohorts, where two of the cohorts were used for the multicenter study and the remaining one was used for the single-center study. The CT scans (taken before and after being classified to be in a severe/critical state and after) were all down-sampled to 64×64×64 image tensors and the image values clamped to the range [−1250, 250]. They were also put through data augmentation such as random horizontal flips and rotations. Each CT scan was then encoded into a 128-dimensional feature vector by 3D ResNet. Clinical data for the patients were put into a 61-dimensional vector which is processed by a multilayer perceptron with identity connections. Long Short-Term Memory (LSTM) was used for its high capacity in modeling large amounts of information and combining clinical features and CT scan features using concatenation. The LSTM was built by a single-layer network with an embedding dimension of 189 and 378 hidden features. The output was a 378×7 tensor normalized with a SoftMax layer which can be interpreted as the probability of the patient being classified as severe/critical. For single center experiments, the model was trained with a learning rate of 0.05, a batch size of 32 and 100 epochs all using the Adam optimizer. For multicenter experiments, the model was also trained on an Adam optimized but at a fixed learning rate of 0.01 for 50 epochs. On comparing seven different models, the authors reported the highest accuracy for their proposed model at 87.7%.

The purpose of the research in [128] was to introduce a machine learning algorithm to predict future intubation durations of patients who were diagnosed or suspected to have COVID-19. The need for this model is due to the scarcity of ventilators and intubation machines, so physicians in emergency departments and the medical industry can better allocate resources based on the likelihood of patients needing them for survival. Data was collected from 4078 COVID-19 patients. The mean age was 58.6 years old. In total, 65.4% of the patients were female and 11.03% were intubated. Of the patients, 24.9% died. As the model was used for predicting intubations, positive labels for intubation for patients were placed after a patient was intubated for 72 h from the end of a 24-h sampling window. Patients who were deceased within the 72-h period after the first 24-h sampling were also listed with positive labels. Data was sampled to a 30 min resolution using feed-forward imputation. In total, 451 patients were intubated for a stay of 6.7–8.4 days. Using 24-h sampling windows, the prediction status was trained using 72 h from the end of the sampling window with continuous risk predictions every 12 h from admission. Comorbidity and time series data were used in a random forest classifier. The class weight balancing was optimized based on the training and validation datasets. The model predictions were defined as positive alert prior to intubation, positive alert with no intubation, no alert with intubation, and no alert with no intubation. Risk assessments were done with the model every 12 h and newly generated risks were used to update risk assessments for the patients. Model performance was compared to the ROX index, which is a ratio for measuring oxygen saturation, respiratory rate, and a fraction of inspired oxygen to predict progression to intubation. The AUC for the ROX index was 0.64 whereas the AUC for the model was 0.84, which is significantly better.

A study that evaluates laboratory data and mortality from patients who tested positive for COVID-19 was reported in [129]. A machine learning model using five serum chemistry parameters to predict death up to 48 h prior to patient death was developed with the objective to identify which prognostic serum biomarkers contributed to the greatest risk of mortality. It also examined the impacts of features that are combined to form different models and their predictions and showed the values that have the highest impact on predictions. The dataset was obtained from a total of 398 patients of which 43 patients were deceased. A total of 26 serum chemistry and blood gas laboratory parameters were collected from patients who had the values captured within 48 h of their death. From the 26 values, the multivariate feature imputation handled missing values and identified potential weights from the features. It then categorized the features into the five most significant features. The five serum chemistry laboratory parameters that were identified by logistic regression were c-reactive protein, blood urea nitrogen, serum calcium, serum albumin, and lactic acid. Using these five main features, an SVM was trained and Shapley values were calculated for the features used in profiling the influence of the features in the model’s prediction. The model was then compared to other prognostic models such as APACHE II, SOFA and CURB65 which have AUCs of 0.84–0.96. The proposed SVM model had a 91% sensitivity and 91% specificity and an AUC of 0.93 and an AUPRC of 0.76. The positive predictive value of predicting the risk of mortality in COVID-19 patients at least 48 h before death was 62.5%. It was discovered using the Shapley values that the model has three main influencing features: C-reactive protein, lactic acid and serum calcium.

A model developed for predicting the outcomes of COVID-19 patients who had cancer-specific risk factors was presented in [130]. The model predicted the level of oxygen needed in cancer patients who tested positive for COVID-19 using clinical variables from both pre- and post-positive diagnoses. The dataset consisted of 348 patients, of which, there were 71 severe-early and severe-late patients, and the remaining 206 were considered non-severe. For each patient, 267 clinical variables were extracted. In total, 26 of them were cancer-related variables, 195 were preexisting diagnosis-related variables, 27 were clinical laboratory variables, 13 were radiology variables, and 6 were basic patient variables. The top variables from these categories were identified to reduce overfitting and minimize bias within the models. Several models were developed with different variable combinations for getting the highest accuracy. Clinical variables used in the model were those that were collected before or at the time of the patient being diagnosed with COVID-19. A random forest ensemble machine learning algorithm, consisting of multiple independent classifiers was used for training the model. Each classifier was trained using a different subset of training variables. The random forest model consisted of 500 decision trees and each with a maximum depth of 10 nodes and a minimum of 1 sample per leaf. The model was evaluated using 10-fold stratified cross-validation, with 10% of the data being used for the validation and the remaining 90% used for training. The process of training and validation was repeated 10 times and each subject was assigned to the test set only once. The model was able to predict the severity of COVID-19 symptoms in cancer patients well with an AUC of 70 to 85%. It had a particularly high performance in identifying patients who were at high risk and needed oxygen within the next three days. Table 5 summarizes the results from these prognostic models identified in this research.

### 5.2. Longitudinal Models

The study conducted in [131] used lung opacification as a quantitative measurement of CT imaging to monitor COVID-19 disease progression. A commercially available deep learning model was used for initial CT image analysis and compared to follow-up CT scans. Patients were split into four different clinical types based on disease severity which were mild, moderate, severe, and critical. The lung opacification percentages showed significant differences between the initial clinical types at the baseline CT imaging and lung opacification percentages increased at the first follow-up. Research in [132] showed a small case study to examine the use of a deep learning model for CT segmentation and quantification of lung opacities in COVID-19. A CNN based on U-Net architecture was used to monitor disease progression in two cases over a period of time. In [133], the study proposed a deep learning model capable of incorporating longitudinal CXR imaging for time-to-event analysis. A convolutional LSTM and RNN-LSTM were used for image feature extraction, which was then concatenated with time-fixed variables. This vector was then passed through a set of fully connected layers to give a prediction of risk. A concordance error of 20% was found for the prediction of hospital admission assessment. The study in [134] used a longitudinal dataset to enhance their deep learning model for better disease segmentation and progression tracking for COVID-19 patients. The dataset had CT imaging from 38 patients with multiple CT scans that could be used for the longitudinal model. Results showed that the proposed longitudinal network had higher Dice Similarity Coefficients than the static network.

Research in [135] compared a deep learning model used for the quantification of serial CT imaging with a standard quantitative CT scoring system for the assessment of COVID-19 patients. A total of 95 patients with 465 serial CT scans were scored and a correlation was examined between the two different methods. Results showed a Spearman Coefficient of 0.920 between the groups thus confirming the potential for disease severity assessment of COVID-19. In [136], the goal of the study was to develop a deep learning model to quantify lung infiltrates in COVID-19 CT imaging. In total, 24 patients were used along with 72 serial CT scans to test the model. The proposed deep learning model was based on the UNet framework for automatic segmentation, elastic registration between the two serial scans, identification of affected lung regions, and quantification of disease progression. The results showed between model detection and manual identification a Dice coefficient of 81%. For the detection of pneumonia regions, the model had a sensitivity of 95% with a specificity of 84%. The study in [137] developed a deep learning model whose primary goal was to track the disease progression of COVID-19 through the use of longitudinal CT imaging. FC-DenseNet56 was used to process slices and a 2.5D model was for the two scan time points. This was compared to a static version of their model that did not incorporate previous scan input data. The results showed that the new proposed longitudinal model had a superior performance for the segmentation of CT scans.

The study in [138] proposed a model that uses Tchebichef moments to develop an objective quantification assessment for COVID-19 CT imaging. Blur change observations were captured using the Tchebichef model. Subjective scores were obtained on the CT imaging from radiologists and compared to the Tchebichef model’s objective score. The results showed a correlation between the two groups. The study in [139] used machine learning to assess COVID-19 disease progression based on longitudinal measurements of eleven clinical features. A total of 144 patients were used along with 3065 readings for 124 different types of measurements over a period of 52 days. These were used to construct a classifier for COVID-19 severity prediction. The results for the model showed a sensitivity of 0.70, a specificity of 0.99, a positive predictive value of 0.93, and a negative predictive value of 0.93.

The study in [140], examined the challenges associated with acquiring training data for deep learning methods for longitudinal studies. They proposed a new self-supervised learning method to overcome this in the quantification of COVID-19 infections in CT imaging. Fully convolutional DenseNet was used as a framework for their longitudinal model and concatenation of 2D CT slices was used as input. Two different data augmentation methods were utilized which included context disordering and black patches. An independent dataset was used for training and validation that consisted of 37 COVID-19 positive patients while the testing dataset consisted of 25 COVID-19 positive patients with 50 longitudinal CTs. The longitudinal classification was conducted to predict the presence of ground glass opacities and consolidation. The results showed Dice scores of 62.3%, 63.5% and 63.6% for static segmentation with no pretraining, black patches, and context disordering, respectively. For longitudinal segmentation, the Dice scores were 64.4%, 65.2% and 65.9% for no pretraining, black patches, and context disordering, respectively. Context disordering performed best for longitudinal classification with an AUC of 0.844 and an accuracy of 0.755. The study in [141] analyzed the longitudinal comparisons between COVID-19 and mycoplasma pneumonia CT imaging using a deep learning model. The dataset included 10 patients positive for COVID-19 and 13 patients with mycoplasma pneumonia with CT imaging and clinical information available. The Pneumonia-CT-LKM-PP deep learning model was deployed for analysis of the number, volume and involvement of lobes for pulmonary lesions. The changes over time were then assessed in three subsequent CTs. The results showed no difference in lymphocytes, but C-reactive protein was significantly higher in those with mycoplasma pneumonia. In terms of CT imaging, the number, volume and involvement of lobes peaked for COVID-19 around 7–14 days after infection while for mycoplasma pneumonia there was no peak or declining trend over time.

Researchers in [142] developed multiple deep learning models based on longitudinal CXR imaging and clinical variables to predict COVID-19 patient mortality using a longitudinal transformer-based network (LTBN). The dataset consisted of 654 COVID-19 patients with 5645 total CXR images. One institution was used for training the models while the other institution was used for testing the models and pairwise comparison for performance. Five models were evaluated which included longitudinal CXR before ICU admission, longitudinal CXR during ICU admission, all longitudinal CXR, clinical and laboratory data at ICU admission, and a combination of longitudinal and clinical variables. One of the models based on all longitudinal CXR had an AUC of 0.702 and an accuracy of 0.694 while only clinical data had an AUC of 0.653 and an accuracy of 0.657. A combination of both longitudinal and clinical data yielded a performance increase to an AUC of 0.727 and an accuracy of 0.732. The study in [143] examined the progression of COVID-19 severity over time using longitudinal CT and clinical data. The data set consisted of 119 COVID-19 patients with 341 longitudinal CT scans from the Shanghai Public Health Clinical Center. This group was subdivided into 90 patients with 273 CT scans who were discharged and 29 patients with 68 CT scans who developed more severe COVID-19 symptoms or died. A deep learning model called BCL-Net was used to extract features from CT imaging which utilized four additional multicenter datasets in order to develop the image segmentation model. An ensemble learning method was used to predict COVID-19 severity based on CT imaging and clinical features. The combination had an AUC of 0.900 in the five-fold cross-validation compared to an AUC of 0.857 and 0.844 for CT and clinical variables only, respectively. The longitudinal model was divided into six groups separated by three days and had AUCs of 0.688, 0.833, 0.901, 0.942, and 0.946 for a combination of CT and clinical data which again outperformed both CT and clinical data alone.

This study [144] used a deep learning model for CT image segmentation and prognosis analysis. The dataset that was utilized was from 369 non-contrast CT examinations that were positive for COVID-19. CT image segmentation was conducted, and specific biomarkers were extracted for prognostic analysis, which were the total opacity ratio (TOR) and consolidation ratio (CR) based on network predictions. Three of the five datasets were selected for prognosis in which both TOR and CR, along with patient demographic and laboratory test information were used. For Institute-1 and Institute-2, the adverse outcome was ICU admission or death, while Institute-3 was mechanical ventilation (MV) usage or death. A generalized linear model was applied for each dataset group and the following AUC with 95% CI was found. Institute-1 had an AUC value of 0.85(0.77, 0.92), Institute-2 had an AUC value of 0.93(0.87, 0.98) and Institute-3 had an AUC value of 0.86(0.75, 0.94). The chosen predictors were age, white blood cells, and platelets for two groups, while oxygen saturation was the chosen predictor for another group. Authors in [145] developed a convolutional neural network to classify CT images from COVID-19 positive vs. non-COVID-19 pneumonia. The dataset to train, validate and test the model was obtained from 13 different international institutions. The model was then used on follow-up scans to track features over time and predict COVID-19 severity over time. The results showed AUCs and accuracies above 0.80 for 7 of the 13 institutions.

The study in [146] analyzed the prognostic capabilities of radiomics modeling in CT-based imaging. A dataset of CT scans was curated from 24,478 patients and was reduced to 14,339 patients. A deep learning model was deployed for whole lung segmentation and 107 radiomic features were extracted. A combination of four feature selection algorithms and seven classifiers was applied to determine which model performed best. Ten different strategies were then used to test the model’s prediction for patient survivability. The highest performing combination was ANOVA feature selector with Random Forest classifier with an AUC, sensitivity and specificity of 0.83, 0.81, and 0.72, respectively, in the CT + RT-PCR COVID-19 positive cases. In the ComBat harmonization dataset, the relief feature selector and random forest classifier were the highest performing with an AUC, sensitivity and specificity of 0.83, 0.77 and 0.74. Authors in [121] developed a deep learning model that combined both clinical and biological variables along with CT imaging to predict COVID-19 severity. The dataset was taken from two French hospitals consisting of 1003 patients. Clinical and biological features were assessed to determine their significance in severity which included age, sex, oxygen, saturation, diastolic pressure, respiratory rate, chronic kidney disease, hypertension, lactate dehydrogenase and urea, C-reactive protein, polynuclear neutrophil and leukocytes. Three significant features were found to have predictive value in CT imaging which included crazy-paving lesions and extent of disease, and peripheral distribution of lesions. An AI-severity score was constructed using age, sex, oxygen saturation, urea, and platelet counts to supplement the CT imaging prognostic modeling and was found to outperform 11 previously published mortality scoring systems with a mean difference in the AUC between 0.05 and 0.16.

The study in [147], deployed a hybrid model approach for mortality prediction that included a 3D-ResNet10-based deep learning model along with a 3D radiomics model. The dataset that was utilized was from four institutions that included 400 patients, 346 discharged and 54 deceased that had underlying health conditions such as diabetes, hypertension and cardiovascular diseases. The hybrid model achieved an AUC value of 0.876 in the test set and 0.864 for an external test set. Additional prognostic modeling capabilities were tested through risk stratification ability through classification into a high-risk group with a mean survival time of 13.5 days or a low-risk group with a mean survival time of 23 days. Authors in [148] utilized a CNN model to assess COVID-19 severity in patients. The dataset that was used was MosMed which contained 1110 CT scans. A severity score was used going from 0, 0.25, 0.50, 0.75, and 1.00 which corresponds to absent, mild, moderate, severe and critical COVID-19 infection corresponding to the percentage of pulmonary parenchymal involvement for each patient. K-Fold cross-validation was used to evaluate the model’s performance on the five-fold framework. The average R2 score was 0.843 and a mean absolute error of 0.133. The model was further evaluated for each of the severity classes excluding critical with a recall score of 94%, 96%, 92% and 88% for absent, mild, moderate and severe, respectively. Infection localization was also analyzed and precision of 84.8% with a recall of 75.9% was reported. This study [149] focused on the development of a deep learning model that can extract features from CT imaging in COVID-19 patients and assess disease severity. The dataset was from multiple sources that consisted of 349 CT images from 216 COVID-19 positive patients and 397 COVID-19 negative CT images. The proposed model for COVID-19 detection achieved an AUC of 95.6% with an accuracy of 89.3% and an F1 score of 88.7%. For the prognostic model, the total accuracy was 47%.

Authors in [150] developed a deep learning model to quantify COVID-19 pulmonary lesions from CT imaging. The dataset consisted of only one institution with 144 patients for training and 30 patients for testing the model. Manual ground-truth segmentation was done to assess the accuracy of the automatic segmentation of the model. The prognostic value of the model was also assessed using multivariable logistic regression on several outcomes which include ICU admission, death, hospitalization greater than 10 days, or oxygen therapy. The results for the segmentation showed a Dice coefficient of 0.75 for lesion segmentation. Three different models were explored for assessing the prognosis where model A included clinical variables, model B treated their CT severity score as an independent variable and model C included the lesion quantification from their segmentation model. The C-statistic outcome was 0.82 for model A, 0.89 for model B, and 0.90 for model C. Authors in [151] developed a deep learning model that combines clinical variables with CT segmentation for diagnosis and prognosis of COVID-19 patients. The dataset that was examined included 839 COVID-19 positive patients, 874 patients with viral pneumonia, and 758 normal patients. Further analysis of their prognostic model was done on another dataset from Stony Brook University on 288 COVID-19 positive patients. The first method included using an intensity map projection to create a 2D representation from the CT imaging. This was compared with the multitask segmentation model that extracted four different pulmonary lesion types. Combinations of these methods were analyzed on both datasets for predicting patient mortality. For the first dataset, the top performing model was a combination of the patient demographics, top three PCA features from the CT classifier and the segmentation features for an AUC of 0.80. For the Stony Brook dataset, the same combination model was the highest performing with an AUC score of 0.81.

The study in [152] developed a framework for the detection of COVID-19 and prediction of ICU admission for patients. The method involves two phases called Early Diagnostic Phase (EDP) and Early Prognostic Phase (EPP). The dataset that was used contained 159 COVID-19 positive patients and 376 normal patient CT image. For the EDP, seven different CNNs were trained on the dataset, and data augmentation techniques along with generative adversarial networks were used to prevent overfitting. The highest performing model with no augmentation was EfficientNetB7 with an accuracy of 99.61%. With augmentation, the highest performing models were MobileNetV1 and VGG-16 with 99.57% and 99.14% respectively. For the EPP, lab results were extracted from patients and 25 different machine learning algorithms were applied to these variables. The highest performing algorithms were Ensemble Bagged Trees and Tree (Fine, Medium, and Coarse) with 98.70% and 97.40% respectively. The study in [116] used machine learning algorithms to analyze clinical and laboratory variables along with a CT severity score to predict patient prognosis for COVID-19. The primary goal was to determine the role that this CT severity score has for prognostic value in patient assessment. The dataset included 57 COVID-19 positive patients and 21 clinical and laboratory features were selected. The top 10 predictors were selected which included neutrophil-lymphocyte ratio, fever, prothrombin time, procalcitonin, cough, fibrin degradation products, hemoglobin, differential neutrophil count, differential lymphocyte count, and ferritin. Five features were manually chosen to train the models with the best performance coming from linear SVM with an AUC of 0.69 and 73.7% accuracy.

This study [153] used a deep learning model to extract features from CT images to predict specific outcomes for COVID-19 patients. Two datasets were used for deep learning feature extraction which included the publicly available data from the Lung CT Segmentation challenge on the Cancer Imaging Archive and the National Health Service UK High-Resolution CT images. The dataset that was used to test patient outcomes was from ChelWest hospitals which included 2710 patients. Nine different image features were extracted along with using patient age and sex which were used as inputs for machine learning classifiers. Logistic regression, SVM, random forest, ada-boost, and XGboost were all used to model patient outcomes. The results showed that SVM and random forest performed best for predicting ICU admission with an AUC of 0.77. Logistic regression performed best for predicting mortality with an AUC of 0.73. The study in [154] examined different machine learning techniques to predict COVID-19 severity. The dataset consisted of 287 patients of which 36.6% were classified as severe and 63.4% as non-severe based on a previous literature’s severity assessment. In total, 23 features were selected including one clinical, one CT, and 21 laboratories which were then used in three models that included a support vector machine, random forest, and logistic regression. Random forest performed the best with an AUC of 0.97 for predicting COVID-19 severity compared to 0.948 for SVM and 0.928 for logistic regression. Feature importance was further analyzed and the top three were reported to be the CT feature, neutrophil to lymphocyte ratio and lactate dehydrogenase. The study in [155] developed a machine learning model that utilized clinical variables along with CT features to predict the need for mechanical ventilation in COVID-19 patients. The dataset that was used to train their segmentation models was the COVID-19 Lung CT Lesion Segmentation challenge. CT Lung segmentation was performed with Bi-directional Conv-LSTM UNet with Densely Connected Convolutions and was measured with the dice coefficient. The dataset that was used for developing the predictive model was from a hospital institution in India that consisted of 302 COVID-19 patients. The two machine learning techniques that were utilized were random forest and XGboost. Three models were created that included clinical parameters alone, clinical parameters with manually derived CT severity scores from a radiologist, and clinical parameters from automated segmentation with the results showing AUC scores of 0.87, 0.89, and 0.91, respectively.

In [156], the authors, used a CNN with autoencoders (AE) for data augmentation to predict COVID-19 patient survival from clinical information and CT images. The dataset contained 320 patients of which 300 were recovered and 20 were deceased from the Tehran Omid hospital. The imbalanced dataset utilized the autoencoder to build reconstructed deceased samples. The results showed that the CNN-AE had an accuracy of 96.05% which was higher than the CNN alone at 92.49%. This was compared with previous existing mortality prediction models that were trained on the same dataset. Research in [157], developed a deep learning convolutional neural network to accurately predict COVID-19 infection. The model included three convolutional layers, three max-pooling layers, and one fully connected layer with an RMSprop optimizer to analyze CT imaging. Data preprocessing was conducted through image resizing, image normalization, and data augmentation. The dataset that was utilized was an open source COVID-19 dataset from Kaggle that contained 1252 COVID-19 positive CT images and 1240 normal CT images. The proposed model had a precision score of 0.96, recall of 0.96, and F1 score of 0.96 which outperformed other pretrained models such as VGG-16, DenseNet121, etc. The accuracy of the model was 95.78%, with a sensitivity of 96% and a specificity of 95.56%.

Another longitudinal study took place at Charité University Hospital, Berlin, Germany [158] in which 86 different diagnostic variables over 687 sample points were collected from 139 COVID-19 patients. The authors used machine learning in conjunction with their data order to determine therapeutic needs for patients and to predict recovery time as well as the risk of deterioration for patients. The algorithm was based on gradient-boosted trees to build a predictive model which leveraged proteome and other clinical data to predict whether a ventilator would be necessary for treatment. The models that incorporated the proteomic and clinical data had much higher predictive scores (AUROC = 0.99) than predictive models based on typical COVID-19 risk factors or molecular predictors. To test the model further, it was deployed on an independent sample of patients from another hospital with COVID-19. When applied to these patients, the model also performed well in predicting (AUROC = 0.97) the need for ventilation. The machine learning models also predicted the WHO severity grade based on proteomic and clinical data. The research team was also able to identify 11 proteins and nine clinical lab markers as predictors (FDR < 0.05) that the patient’s condition would decline, as well as proteins and other variables that were predictive of the remaining time in the hospital. The authors also developed a machine learning model that predicted WHO grade severity based just on the first time point data. They found that the predictions from this model correlated with the time remaining in the hospital. A summary of this section is shown in Table 6.

## 6. Conclusions

In this review paper, a summary of research pertaining to COVID-19 diagnosis has been presented with the goal of assessing the importance of machine learning in this domain. Several open-source datasets used for analysis were also presented which mostly consisted of CT and CXR. It is evident that machine learning algorithms especially deep learning has made a tremendous contribution to all aspects of patient outcome. Furthermore, in the majority of the papers, CT imaging proved to be extremely crucial in the decision-making process. As with preceding review papers, there was also a concern about the applicability of some diagnostic models due to the complexity of the proposed approach and some research provided counter-intuitive evidence of light-weight models being as accurate in diagnosing COVID-19. Several research papers presenting multiple deep learning models appeared to be focused on model comparison lacking the feasibility of deployment in the form of a web-based API to facilitate information exchange. Prognostic factors identified in longitudinal studies were focused mostly on hospitals and institutions. As a result, they were very crucial to enhancing patient care and managing already stressed hospital resources. These studies mostly presented simpler machine learning models with significant accuracy. However, the majority of the datasets used in their research were not freely available due to restrictions unlike CT scans used for diagnostic purposes. There was no centralized repository to rapidly share developed models with other hospitals. COVID-19 has identified the need for an integrated infrastructure that not only allows clinicians to share research findings but also provides an avenue to share their deployed machine learning model, while hospital infrastructure may vary significantly, patient bio-markers are similar making this globalized solution achievable. 

## Figures and Tables

**Figure 1 diagnostics-12-01853-f001:**
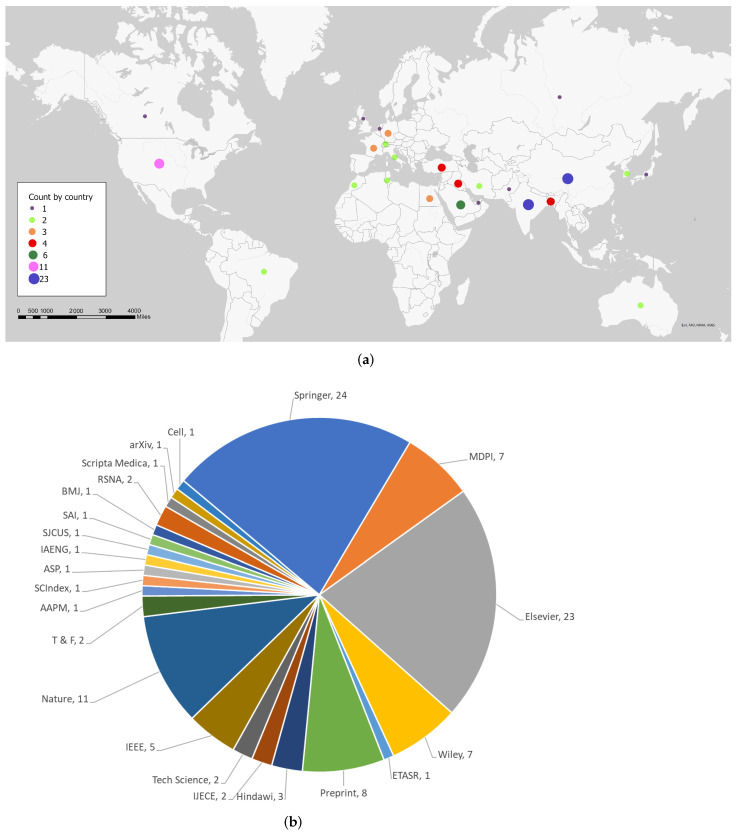
Research presented in this paper summarized based on (**a**) country and (**b**) publisher.

**Figure 2 diagnostics-12-01853-f002:**
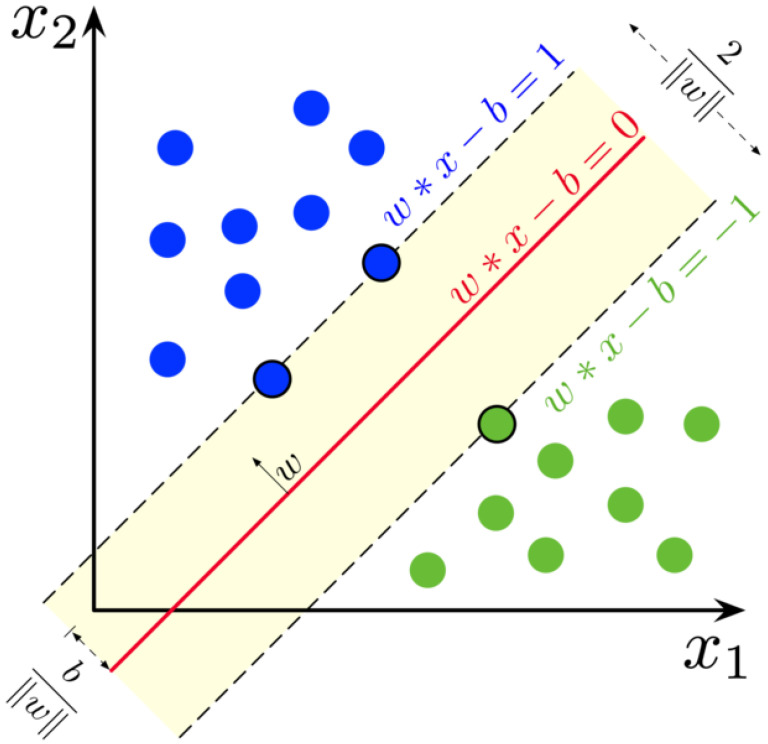
Support vectors on yellow area, are used to create boundaries also known as hyperplanes. The SVM in the figure is using a linear kernel denoted by the straight line equation. Image distributed under a CC BY-SA 4.0 license.

**Figure 3 diagnostics-12-01853-f003:**
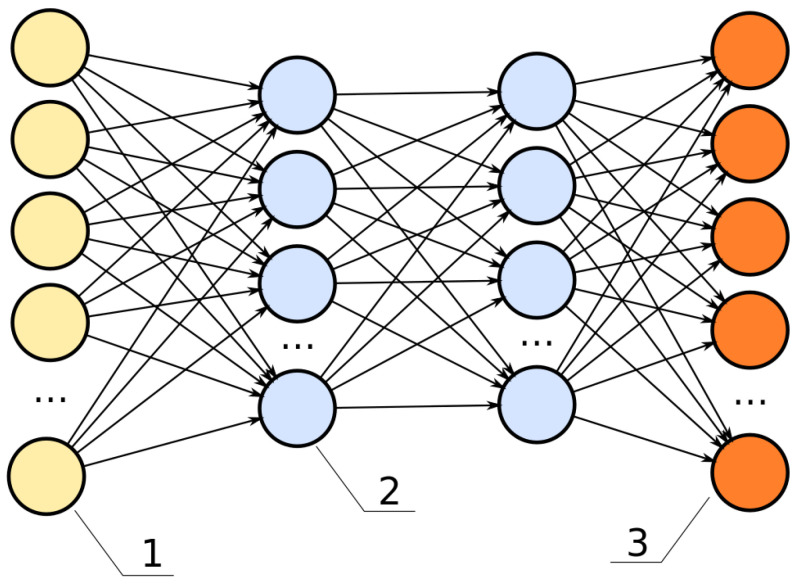
The structure of DNN model. Image distributed under a CC BY-SA 4.0 license. Number one represents input layer followed by two which are the hidden layers and finally three which is the output layer.

**Figure 4 diagnostics-12-01853-f004:**
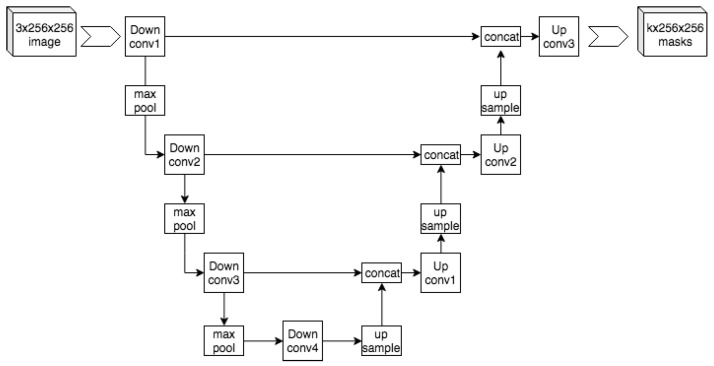
The structure of a UNet model similar to the original one proposed in [17]. Image distributed under a CC BY-SA 4.0 license.

**Figure 5 diagnostics-12-01853-f005:**
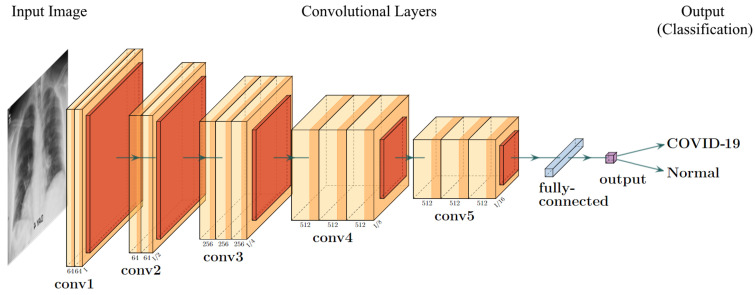
The structure of VGG-16 model adapted for prediction of COVID-19 using CXR.

**Figure 6 diagnostics-12-01853-f006:**
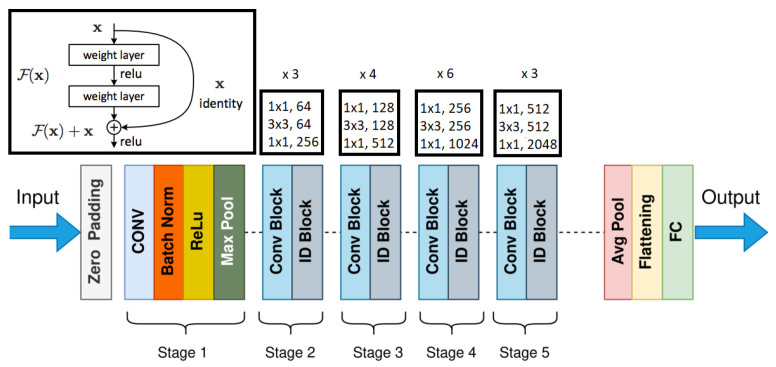
The structure of ResNet 50 model [19], distributed under a CC BY-SA 4.0 license.

**Figure 7 diagnostics-12-01853-f007:**
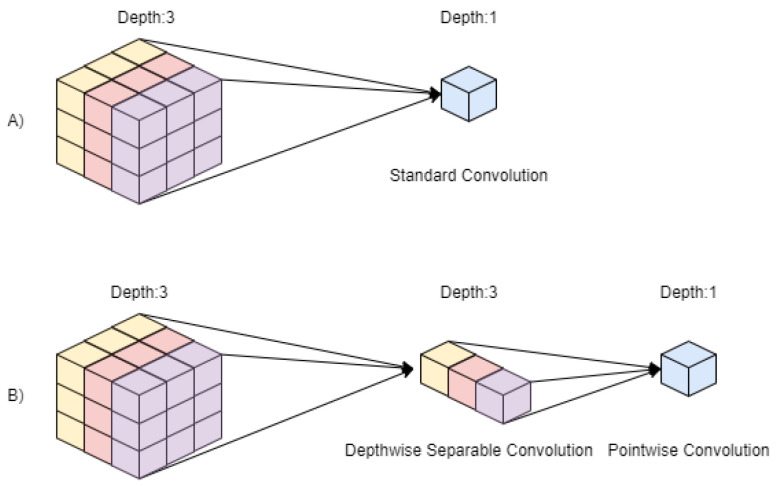
A visual interpretation of (**A**) standard convolution compared to (**B**) depthwise separable convolutions used in MobileNet.

**Figure 8 diagnostics-12-01853-f008:**
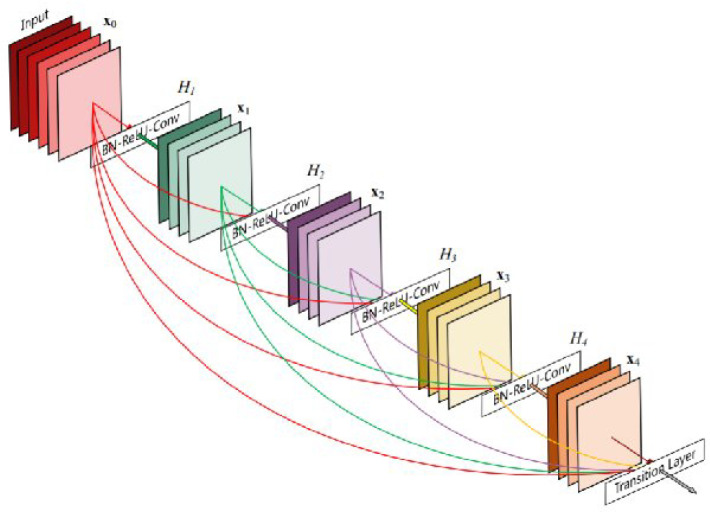
The structure of DenseNet [21] block shown in [22]. Features are added from all preceding blocks.

**Figure 9 diagnostics-12-01853-f009:**
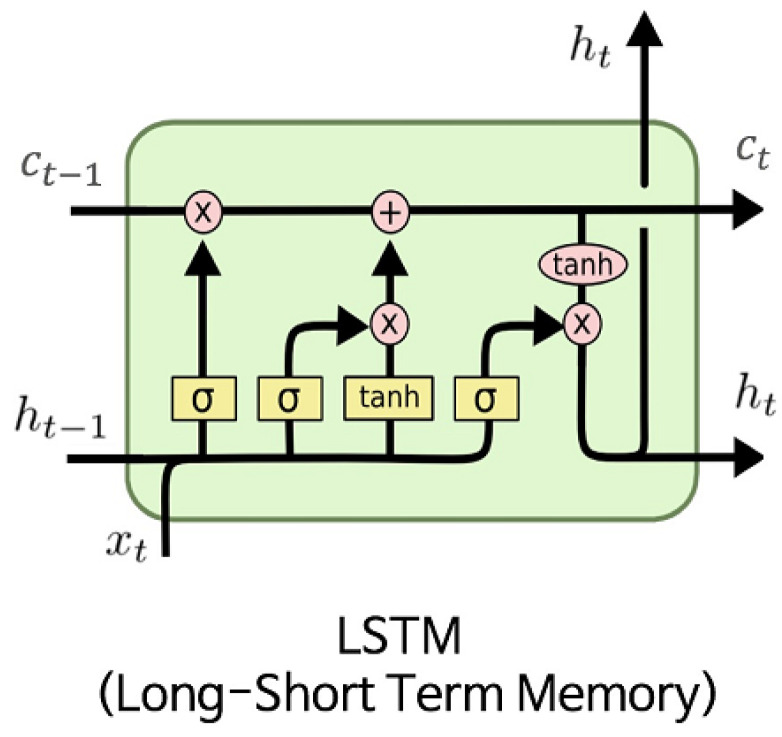
The structure of an LSTM block used to process longitudinal data and distributed under a CC BY-SA 4.0 license.

**Figure 10 diagnostics-12-01853-f010:**
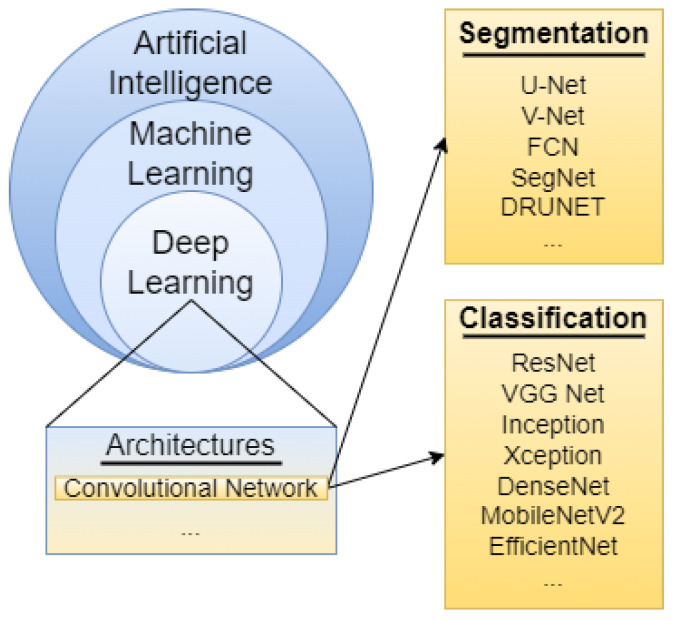
Common deep learning architectures explored in COVID-19 diagnosis.

**Figure 11 diagnostics-12-01853-f011:**
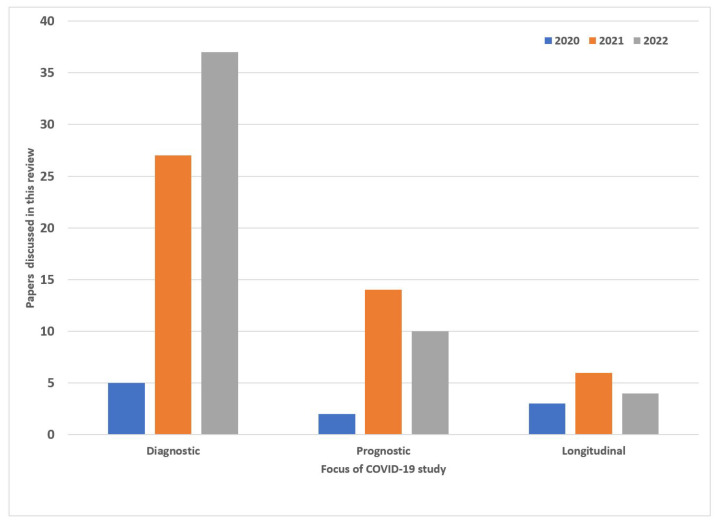
Papers reviewed in this research presented based on the three primary objectives.

**Table 1 diagnostics-12-01853-t001:** Publicly available COVID-19 CT scan datasets.

Dataset Name		Modality	Size of Dataset	Link to Dataset
	1a	CT	120	[25]
RICORD	1b	CT	120	
	1c	CXR	998	
COVIDx CT	2a	CT	194,922	[26]
	2b	CT	201,103	
Extensive COVID-19 chest X-ray and CT chest images dataset		CXR	9544	[27]
		CT	8055	
MosMed		CT	1000	[28]
COVID-CT		CT	812	[29]
China Consortium of Chest CT Image Investigation (CC-CCII)		CT	514,103	[30]
COVID-19 CT segmentation dataset		CT	929	[31]
COVID-19 CT lung and infection segmentation dataset		CT	20	[32]
CoronaHack chest X-ray dataset		CXR	5910	[33,34]
COVIDx CT-2A		CT	194,922	[26]
COVID-19 radiography database		CXR	21,165	[35,36]
COVID-19 image data collection		CXR	1000	[34]
Augmented COVID-19 X-ray images dataset		CXR	1824	[37]
SARS-CoV-2 CT scan dataset		CT	2482	[38]
COVID-CT set		CT	63,849	[39]
Chest X-Ray images (Pneumonia)		CXR	5863	[40]
COVID-19-vs-normal dataset		CXR	800	[41]
COVIDx-US		LUS	12,943 + 150 videos	[42]
COVID-19 patients lungs X-ray images 10,000		CXR	3428	[34,40,43]
COVID-19 & normal posteroanterior (PA) X-ray		CXR	280	[44]
COVID-19 image dataset		CXR	317	[45]
COVID-19 X-ray classifier		CXR	407	[34,40]
COVID-19-CT-Seg		CT	20	[32]
COVID-19 detection X-ray dataset		CXR	3071	[34,46,47]
Chest X-ray (COVID-19 & Pneumonia)		CXR	6432	[34,40,48]
SegmentedLungCXRs		CXR	342	[49]
Actualmed COVID-19 chest X-ray dataset initiative		CXR	238	[50]
COVID-19 X-ray dataset (Train & Test Sets)		CXR	188	[34,51,52]

**Table 2 diagnostics-12-01853-t002:** Machine learning approach summarized using CXR data.

Paper	Year	Model	Goals	Accuracy
[53]	2022 January	CovNNet	Image augmentation generating hybrid deep learning model for classification.	80.90%
[54]	2022 January	Join-fusion model using EfficientNetB7	Combine clinical and CXR for binary prediction	97%
[55]	2022 January	DNN + CNN	Model does binary classification followed by segmentation	94.60%
[56]	2021 January	Hybrid model developed from 11 different CNN architectures	Binary prediction	98.54%
[57]	2021 December	XR-CAPS	Feature extraction and segmentation by using foundations of UNet	93.20%
[41]	2020 November	COVID-CheXNet	Contrast enhancement on the CXR and noise reduction techniques prior to classification	99.99%
[58]	2022 January	Deep CP-CXR	Binary classification as well as COVID-19, normal, and pneumonia	98.57%
[59]	2022 May	CapsNet	Lower-level capsules reuse features for image classification.	89.87% avg.
[60]	2022 May	Model comparison with ResNet18 producing highest accuracy	Classify healthy, COVID-19, or other viral-induced pneumonia	97%
[61]	2022 May	EfficientNet B5 transfer learning.	COVID-19, pneumonia, or normal classification	86.67%
[62]	2022 May	Feature maps utilized sequentially with MobileNetV2, EfficientNet0, and DarkNet53.	COVID-19, viral pneumonia, or normal classification	99.05%
[63]	2022 May	Fusion-based CNN, COVDC-Net	COVID-19, healthy, bacterial pneumonia, and viral pneumonia classification	96.48%
[64]	2022 April	Lightweight CNN edge computing	Image augmentation using DCGAN increase sample size for classification.	87%
[65]	2022 April	Lightweight CNN C-COVIDNet	COVID-19, pneumonia, or normal classification using ROI extraction of lungs.	97.5%
[66]	2022 April	COVID-CXDNetV2 with YOLOv2 and ResNet	Normal, COVID-19, and pneumonia classification	97.9%
[68]	2022 May	Semi-supervised deep learning.	Normal, bacterial pneumonia, COVID-19, lung opacity, and viral pneumonia classification	81%
[69]	2021 December	Traditional CNN	Identification of structural abnormalities in CXR	97.67%
[70]	2022 January	M-qXR commercial model	Risk stratification in COVID-19 diagnosis	98%
[71]	2022 May	VGG16	COVID-19 diagnosis by incorporating image augmentations with fast timing.	99.3%
[72]	2022 April	RESCOVIDTCNnet	Model proposed after comparison with InceptionV3, ResNet50, and ResNet50-TCN	99%
[67]	2022 February	LSTM	Using LSTM architecture for classification of COVID-19 and Influenza from CXR	98%

**Table 3 diagnostics-12-01853-t003:** Machine learning approach summarized using primarily CT data.

Paper	Year	Model	Goals	Accuracy
[73]	2021 February	Multiple model comparison with VGG-19 having best classification accuracy	Classification of covid, normal, and pneumonia across open-source datasets	94.52%
[74]	2021 September			98.87%
[76]	2022 January			90.50%
[75]	2020 November	CoroDet	Used CT and CXR to perform multiple class-wise predictions	91.20%
[77]	2022 February	Developed a hybrid 4-layer CNN after testing eight different models	Proposed rapid real time COVID-19 detection using hybrid approach on CXR and CT	98%
[78]	2021 August	ResNet50	Application on CXR	98%
[79]	2021 July	COVID-Net CT-2 and ResNet-v2 models	Research on both CT and CXR using COVIDx-CT dataset	98.7%
[80]	2021 March	DREnet	Pretrained ResNet 50 model	79% precision
[81]	2021 February	ResNet50	Verified that image augmentation not necessary for COVID-19 detection and utilized compressed JPEG files.	98.8%
[82]	2020 August	ResNet 18,50, and 101 alongside SqueezeNet	Tested model on CT images showed ResNet 18 to have best results.	99.4%
[39]	2021 March	ResNet50V2	Image preprocessing and removal of CT slices not relevant to ROI.	98.49%
[83]	2021 February	Inception-v3 model with patch-based ROI	Tissue delineation using ROI for classification	89.5%
[84]	2022 January	Model comparison InceptionV3, ResNet50, VGG16, and VGG19	Best performance using ResNet for CT scan classification.	98.86%
[85]	2021 June	MobileNet-V2	Purpose was to demonstrate the feasibility of light weight models in analyzing CT scans.	85.6%
[86]	2020 November	COVID-AL	Weakly-supervised COVID-19 detection module based on UNet for segmentation.	96.8%
[87]	2022 January	Pretrained ResNet18, ResNet50, and ResNet101	Explores the feasibility of pre-trained models without providing labelled training data.	99.4% F1
[88]	2021 December	EDECOVID-net	Uses radiomic features to differential between COVID-19 and pulmonary edema	98%
[89]	2021 December	LSTM	Applies LSTM on both CXR and CT images for classification	97.3%
[90]	2021 December	CNN-based approach	Performed binary as well as ternary classification, using CT images	98.79%
[91]	2022 June	DensNet121 (pre-trained)	Transfer-learning deep CNN with image augmentation.	96.52%
[92]	2022 May	Bidirectional LSTM	Classification using CT for normal, viral pneumonia, lung opacity, or COVID-19	99.07%
[93]	2022 May	GAN	Combination of GAN and deep learning increased image data for training	89% AUC
[94]	2022 May	UNet	Segmentation of lung by utilizing contrast limited adaptive histogram equalization (CLAHE)	98%
[95]	2022 May	UNet	Edge detection, thresholding, and UNet are compared using IoU metric for segmentation.	95%
[96]	2022 May	DCML	Multiple preprocessing using Fast AutoAugment, data balancing steps prior to binary classification.	98.7%
[97]	2022 May	DMDF-Net	CT scan segmentation for COVID-19. Fusion of UNet and MobileNetV2.	0.75 DICE
[98]	2022 April	UNet with cross-loss function	Segmentation using image preprocessing with random window width and levelling.	0.83 DICE
[100]	2021 May	VB-Net	automatic segmentation and delineation of affected regions with COVID-19 from CT scans.	-
[101]	2021 July	Deep Covix-net	Image segmentation with texture analysis using GLCM, GLDM, FFT, DWT on CT and CXR.	97%
[99]	2021 February	VGG style CNN	Evaluation of multiple deep learning model using two public and two private datasets.	93%

**Table 4 diagnostics-12-01853-t004:** Machine learning approach summarized using other datasets and ensemble techniques.

Paper	Year	Model	Goals	Accuracy
[102]	2021 July	VGG-16, ResNet50, and Xception	Grouped results for prediction of COVID-19 using CT scans	98.79%
[103]	2021 June	VGG19, ResNet101, DenseNet169, and WideResNet 50-2	Looked at CT and CXRs and compared results with existing models on 5 datasets	99.75%
[104]	2022 January	VGG19, ResNet50, DenseNet201, Random Forest and Extra Trees	Used a Stacking and Weighted Average Ensemble (WAE) approach with these algorithms on two separate datasets	96.65% F1
[105]	2021 July	Meta-classifer, Efficient-Net, PCA, SVM and Random Forest	Model tested on 8055 CT images and 9544 CXR for prediction of COVID-19 and normal	99% precision
[106]	2021 June	Haralick and SVM feature extractor	Binary classification on IKONOS-CT images.	96.99%
[107]	2022 April	Discriminant Analysis, Ensemble, Random Forest, and SVM	Combined ensemble approach with feature fusion for classification.	96.18%
[108]	2021 November	Using Random Forest with first and second order texture analysis.	Four levels of severity of COVID-19	90.95%
[109]	2021 November	Random Forest	Combination of ML and Neural Network algorithms tested using multiple features along with CT images.	91.38%
[110]	2022 May	ResNet-50 and SVM.	Binary classification of COVID-19 in a hospital in Brazil.	98.20%
[111]	2022 May	Feature fusion with DenseNet, ResNet34 and VGG16	Classification of COVID-19, pneumonia, or normal using a dataset of 6518 CXR images	97.3%
[112]	2022 April	VGG16 and SVM	Binary classification using Basrah dataset	99% F1
[113]	2022 May	CDBN, HRNet and VGGNet feature fusion	Lung segmentation and classification.	95%
[114]	2022 April	Ensemble approach by fitting multiple deep learning models	Transfer learning to detect COVID-19 in 1000 CXR images	100% precision
[115]	2021 December	Light Gradient Boosting	Predict COVID-19 ARDS, ICU, mortality risk	79% AUC
[116]	2021 December	SVM	Feature reduction from clinical factors + CT images using multiple machine learning techniques.	73.7%
[117]	2022 April	CNN and RSTN	Binary classification of COVID-19 using lung ultrasound	90.18%
[118]	2021 November	Two-Stream Inflated 3D ConvNet (I3D)	Classify three categories of normal, interstitial abnormalities, and confluent abnormalities using lung ultrasound	86% F1
[119]	2021 August	InceptionV3	Classification of COVID, normal, and pneumonia.	89.1%
[120]	2020 November	VB-Net	COVID-19 severity detection from CT scan.	91.6%
[121]	2021 January	Pretrained EfficientNet -B0 & ResNet50	COVID-19 severity detection based on clinical factors and CT scans	76% AUC

**Table 5 diagnostics-12-01853-t005:** Machine learning approach summarized for developing prognostic models in COVID-19.

Paper	Year	Model	Goals	Accuracy
[122]	2020 April	Survey on prognostic models which identified 51 relevant studies		
[123]	2021 February	Development of COVID-19 severity assessment (COSA) score	Clinical variables used to predict several COVID-19 outcomes	0.96 AUC
[124]	2021 May	Evaluated 18 machine learning algorithms	Negative outcomes prediction of ICU admission and mortality rates.	>0.80 F1
[125]	2021 February	Random forest, extreme gradient boosting and neural networks	Outcome prediction: Death, ICU admission and ventilation	0.92 AUC
[126]	2021 May	CNN with CT scans and gradient boosting COVID-GMIC-DRC	Clinical data used to predict the risk of deterioration	0.786 AUC
[127]	2020 September	3-D ResNet and LSTM	Used clinical data and clinical data and sequences of CXR scans	0.92 AUC
[128]	2020 November	Random-forest classifier	Predict future intubation durations of patients	0.84 AUC
[129]	2021 March	SVM and logistic regression	Identify serum biomarkers that contribute to the greatest risk of mortality.	0.93 AUC
[130]	2021 May	Random Forest	Predicting outcomes of COVID-19 patients who had cancer-specific risk factors	0.829 AUC

**Table 6 diagnostics-12-01853-t006:** Machine learning approach summarized for developing longitudinal models in COVID-19.

Paper	Year	Model	Goals	Accuracy
[131]	2020 March	QCT-PLO	Used lung opacification-based segmentation for monitor COVID-19 progression.	-
[132]	2020 March	UNet		
[133]	2021 August	LSTM and RNN-LSTM	Time-to-event analysis using CXR data.	20% Conc. Error
[134]	2021 September	Hybrid UNet	Used longitudinal segmentation to identify COVID-19 progression from 38 patients CT scans.	0.837 Dice
[135]	2021 January	COVID-Lesion Net	Serial CT for COVID-19 scoring system to identify disease severity	0.920 Spearman
[136]	2020 August	UNet	Assessment of serial CT scans for disease progression and pneumonitis region identification	0.81 Dice
[137]	2021 October	FC-DenseNet56	Disease progression and interactive segmentation of lungs for COVID-19	0.59 Dice
[138]	2021 April	Tchebichef moments (TM)	COVID-19 progression monitoring using CT blur correction with TM	0.7983 Spearman
[139]	2021 June	SVM	Severity prediction using time-series CT imaging + clinical variables from 841 patients	0.89 AUC
[140]	2022 March	DenseNet	Applied image augmentation techniques like context disordering for longitudinal classification of CT images	0.844 AUC
[141]	2022 February	Pneumonia-CT- LKM-PP	Identify number of lobes that had pulmonary lesions from 10 patients	-
[142]	2022 March	LTBN	Longitudinal CXRs used to assess patient mortality from 654 patients	0.727 AUC
[143]	2022 April	BCL-Net and BC-LSTM Module	Extraction of CT features from longitudinal scans followed by clinical data fusion.	0.900 AUC
[144]	2021 February	Dense 3D network with GLM + Feature selection	Total opacity ratio (TOR) and consolidation ratio (CR) for CT and HER for ICU admission	0.93 AUC
[145]	2021 January	Deep COVID DeteCT	3D deep learning classification across 13 institutions to track disease progression	97.4%
[146]	2022 March	Deep learning with feature selection and texture analysis	Lung segmentation using 107 intensity and texture features from 14,339 CT patients	0.83 AUC
[121]	2021 January	Deep learning with clinical feature correlation	Used multiple features with CT for COVID-19 disease progression in two French hospitals	0.88 AUC
[147]	2021 April	3D-ResNet10	Risk stratification for 400 patients in four institutions	0.876 AUC
[148]	2022 March	Cross-channel attention based deep learning	Diagnosing criticality of patients medical condition from CT scans. CNN has a multi-stage analyzer	0.84 $R^{2}$
[149]	2022 March	Siamese Neural Network (SNN)	Severity scores identified using bounding boxes and used for providing future clinical assistance	87.6%
[150]	2022 March	UNet	Used LDCT for predicting death, hospitalization, ICU, and oxygen therapy	0.75 Dice
[151]	2022 February	InceptionResnetV2 + PCA	Prediction of mortality using CT data with clinical variables using CC-CCII dataset	0.80 AUC
[152]	2022 January	EfcientNetB7 + GAN variants	Early detection and early prognosis using CT and lab data from 15 patients	98.7%
[116]	2021 December	Decision tree, SVM, KNN and ensemble classifiers	ML models for prognostic study using CT scans and 21 clinical features of 1431 patients	73.7%
[153]	2022 May	SVM, random forest, and logistic regression	ICU admission and mortality prediction from two datasets and one hospital	0.77 AUC
[154]	2022 February	SVM	COVID-19 severity prediction from 287 patients	0.97 AUC
[155]	2022 January	Bi-directional Conv-LSTM UNet, Random forest and XGboost	Estimating the need for mechanical ventilation in COVID-19 patients	0.91 AUC
[156]	2021 July	CNN-AE	Autoencoders used to address data imbalance and predict patient survival from 320 patients.	96.05%
[157]	2021 February	Covid CT-net	Used a light weight CNN for COVID-19 prognosis	95.78%
[158]	2021 June	Linear regression, XGBoost 1.2.0 and MaxLFQ	Predict therapeutic needs, recovery time and risk of deterioration from 86 diagnostic variables.	0.99 AUC

## Data Availability

Not applicable.

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
