# Peer review of "A Comprehensive Review of Machine Learning Used to Combat COVID-19"

_diagnostics, 2022, doi:10.3390/diagnostics12081853_

Round 1
Reviewer 1 Report
Manuscript Title: A comprehensive review of machine learning used to combat COVID-19
Reviewer Recommendation : Major revision Required
Comments
1. Authors have mentioned that section 2 of this study was focused on the AI-based algorithms which are extensively used in CT-imaging. Are authors only focused on the algorithms which are used in the analysis of CT-images?
2. Review along with Case study with error values can be more beneficial.
2. Is CNN-deep learning analysis only applicable for imaging analysis, apart from this any other along with pro’s and con’s?
3. Under the subheading of machine architecture authors explained well about the architecture of the model, in addition to that, if authors discuss the input and output of each model it could be more informative.
4. Authors have mentioned that the accuracy of CNN is higher (94.6%) than DNN (84.1%), if authors discuss the reason for the higher accuracy of CNN, it could be more informative. Also men
5. Authors have explained the various Deep learning methods which have been used in the analysis of CT-images, if the author adds the information of limitation which is associated with each DL methods for this analysis it could be more informative.
7. Under the subheading diagnosis using chest radiographs, page no.10 line number 292, it was mentioned that five separate models were used for the validation, if authors discuss the models which are used for validation more in detail, it could be more informative.
8. Authors requested to provide information on major algorithms which are implemented in each DL method either in table or in main text.
9. In some cases, the X-ray chest images are not suitable to analyze the prediction of COVID-19; in such cases how the DL methods work on it? Clarification required on this problem,
10. I suggest authors add the pictorial representation of backend algorithms, which could be an added advantage for the readers.
Author Response
Thank you for your suggestions. We have attached a pdf addressing all concerns.

Reviewer 2 Report
This is a timely and well-focused review article, and its publication in this journal seems to be effective and useful.
One point, the authors describe it in the "article" template, but it is a "review" type manuscript, and that is the content.
Author Response
Thank you for your suggestion. We have modified the 'article' to reflect 'review'.
Round 2
Reviewer 1 Report
Revised version can be accepted for publication
Best wishes to the authors